# An updated microphysical model for particle activation in contrails: the role of volatile plume particles

Joel Ponsonby[1], Roger Teoh[1], Bernd Kärcher[2], and Marc E. J. Stettler[1*]

[1]Department of Civil and Environmental Engineering, Imperial College London, London, SW7 2AZ, United Kingdom
[2]Institute for Atmospheric Physics, DLR Oberpfaffenhofen, Weßling, Germany

*Correspondence to*: Marc E. J. Stettler (m.stettler@imperial.ac.uk)

**Abstract**

Global simulations suggest the mean annual contrail-cirrus net radiative forcing is comparable to that of aviation's accumulated $CO_2$ emissions. Currently, these simulations assume non-volatile particulate matter (nvPM) and ambient particles are the only source of condensation nuclei, omitting activation of volatile particulate matter (vPM) formed in the nascent plume. Here, we extend a microphysical model to include vPM and benchmark this against a more advanced parcel model (pyrcel) modified to treat contrail formation. We explore how the apparent emission index (EI) of contrail ice crystals ($AEI_{ice}$) scales with $EI_{nvPM}$, vPM properties, ambient temperature and aircraft/fuel characteristics. We find model agreement within 10-30% in the previously defined "soot-poor" regime. However, discrepancies increase non-linearly (up to 60%) in the "soot-rich" regime, due to differing treatment of droplet growth. Both models predict that in the "soot-poor" regime, $AEI_{ice}$ approaches $10^{16}$ kg$^{-1}$ for low ambient temperatures (< 210 K) and sulphur-rich vPM, which is comparable to estimates in the "soot-rich" regime. Moreover, our sensitivity analyses suggest that the point of transition between the "soot-poor" and "soot-rich" regimes is a dynamic threshold that ranges from $10^{13}$ kg$^{-1}$ – $10^{16}$ kg$^{-1}$ and depends sensitively on ambient temperature and vPM properties, underlining the need for vPM emission characterisation measurements. We suggest that existing contrail simulations omitting vPM activation may underestimate $AEI_{ice}$, especially for flights powered by lean-burn engines. Further, our results imply that under these conditions, $AEI_{ice}$ might be reduced by (i) reducing fuel sulphur content, (ii) minimising organic emissions and/or (iii) avoiding cooler regions of the atmosphere.

# 1. Introduction

The aviation industry contributes approximately 3.5% towards global anthropogenic effective radiative forcing (ERF) (Lee et al., 2021). This share is projected to increase as the sector is comparatively challenging to decarbonise (Grewe et al., 2021), notwithstanding anticipated compound annual growth in global passenger traffic of 3.6% from 2027 – 2043 (Airbus, 2024). Estimates for 2018 indicate that the largest contribution towards net aviation ERF (+100.9 mWm$^{-2}$) is from contrail-cirrus (+57.4 mWm$^{-2}$), with remaining contributions from accumulated $CO_2$ emissions (+34.3 mWm$^{-2}$), followed by $NO_x$ (+17.5 mWm$^{-2}$) (Lee et al., 2021). The sign and magnitude of the contrail-cirrus ERF depend *inter alia* on optical thickness and spatiotemporal coverage (Kärcher, 2018; Schumann et al., 2012), which are determined by ice crystal microphysics and hence the contrail formation pathway.

Provided the aircraft exhaust plume satisfies the Schmidt-Appleman Criterion (SAC), linear contrail formation occurs within 1 s behind the engine exit plane (Schumann, 1996). In the nascent plume, exhaust gases mix with cooler ambient air resulting in a locally water-supersaturated environment. Under these conditions, particles in the plume can activate to form water droplets, which later nucleate ice homogeneously to form contrail ice crystals. Hence, this pathway is regulated by the number concentration of particles able to activate, which is controlled by their particle size distribution characteristics and chemical composition (Petters and Kreidenweis, 2007). If fewer particles are able to activate, the resulting ice crystals will be larger and less numerous (Teoh et al., 2022), which may decrease both the optical thickness and the spatiotemporal coverage of the cloud. Therefore, contrail-cirrus ERF is inherently sensitive to the characteristics of particles in the mixing plume.

Particles within an exhaust plume are either released by the engine (aircraft mode) or entrained during atmospheric mixing (ambient mode). Aircraft mode particles that evaporate (above) below 350 °C (623 K) (Saffaripour et al., 2020) are defined as (non-)volatile particulate matter (n)vPM. The largest vPM particles are formed from chemi-ions, which are generated during the combustion process (Yu and Turco, 1997). These chemi-ions scavenge sulphuric acid-water clusters and/or condensable particulate organic matter (POM) in the cooling plume to generate liquid aerosol particles with a number emission index $EI_{vPM} \sim 10^{17}$ kg$^{-1}$ (Yu et al., 1998). A smaller, more numerous population of vPM is generated from binary homogeneous nucleation of sulphuric acid and water (Kärcher and Fahey, 1997). However, these particles are efficiently scavenged by the larger particles, enhancing their growth depending on available sulphuric acid and/or POM (Kärcher et al., 2000). They are therefore unlikely to compete with the larger population for plume supersaturations, so we consider the larger population as plume vPM within this work. The geometric mean diameter (GMD) and hygroscopicity of plume vPM depends on the nature and availability of condensable gases in the aircraft exhaust (Kärcher et al., 2000); $GMD_{vPM}$ is typically < 5 nm (Schumann et al., 2002) prior to contrail formation. Depending on the engine architecture, (semi-)vPM lubrication oil may also be released as larger droplets or gaseous emissions, with the latter capable of either nucleating new particles or contributing towards the condensable organic fraction (Decker et al., 2024; Ungeheuer et al., 2022). By contrast, $EI_{nvPM}$ and $GMD_{nvPM}$ do not change in young exhaust plumes, besides a minor increase to $GMD_{nvPM}$ by accretion of condensable material. Under cruise

thrust settings, $GMD_{nvPM} \sim 35$ nm (Durdina et al., 2024) and exhibits dependence on fuel and combustor type (Moore et al., 2017; Voigt et al., 2021).

$EI_{nvPM}$ varies by several orders of magnitude depending on the fuel and engine type (ICAO, 2024), with most engines operating in the "soot-rich" regime, where the majority of contrail ice crystals form via nvPM particles (Kärcher and Yu, 2009). Larger nvPM particles ($GMD_{nvPM} > GMD_{vPM}$) activate preferentially in the plume and can suppress water uptake on
plume vPM particles, defining "soot-rich" baseline conditions of approximately $EI_{nvPM} > 10^{15}$ kg$^{-1}$ (Kärcher, 2018). Upon reducing $EI_{nvPM}$, size differences no longer compensate for differences in relative abundance and concentrations of activated nvPM and plume vPM become comparable, characterizing the "transition regime" (Kärcher and Yu, 2009). Here, baseline conditions depend sensitively on the characteristics of all particle species. Recent developments in lean-burn technology have been shown to further reduce $EI_{nvPM}$ into the "soot-poor" regime, where activated number concentrations become independent
of $EI_{nvPM}$ as $EI_{nvPM}$ decreases. "Soot-poor" baseline conditions have been shown to depend on the relative concentration of ambient particles and the characteristics of plume vPM, including $EI_{vPM}$ and $GMD_{vPM}$ (Yu et al., 2024). Nevertheless, while the role of vPM has been evidenced in model simulations of individual contrails, these effects have not yet been incorporated into global mean contrail-cirrus ERF estimates (Bier and Burkhardt, 2022; Teoh et al., 2024).

Generally, two complementary classes of models are used to describe contrail formation (Bier et al., 2022): those that
focus on jet dynamics and those that focus on plume microphysics. The former include 3D large eddy simulations (LES), which provide insight into the spatial heterogeneity within contrail mixing plumes (Lewellen et al., 2014). Inevitably, the high spatial resolution achieved in these simulations is achieved at the expense of detailed microphysical descriptions (Ferreira et al., 2024). To that end, several LES models prescribe water saturation as the critical condition for contrail ice formation (Paoli et al., 2013; Picot et al., 2015) or heterogeneous ice nucleation as the primary formation pathway (Khou et al., 2017, 2015),
both of which have been rejected by *in-situ* observations (Kärcher et al., 2015). More representative microphysical treatment can be achieved using 0D box- and parcel model simulations (Bier et al., 2022; Rojo et al., 2015; Yu et al., 2024). Here, the dilution of a parcel of exhaust air is simulated and microphysical phase transitions such as particle activation and homogeneous ice nucleation are tracked. While these models are unable to incorporate feedback between different plume parcels, which may otherwise lead to a diversity of particle history (Lewellen, 2020), they are configured for sophisticated treatment of complex
ice microphysics, which is critical for describing contrail properties (Yu et al., 2024).

Here, we undertake a literature review of the microphysical pathway to contrail formation, to better understand the role of plume vPM in contrail formation. We then extend two parcel models using detailed microphysics to account for activation of plume vPM. In both models, we prescribe plume vPM properties at the time of droplet formation and ice nucleation (0.1 – 1 s after emission) rather than explicitly modelling their formation. The two models include: (i) a minimal
microphysical framework developed by Kärcher et al (Kärcher et al., 2015), henceforth referred to using the shorthand K15 and (ii) a more complex numerical parcel model (pyrcel) developed by Rothenberg and Wang (Rothenberg and Wang, 2016). The extended K15 model is designed for future integration into global contrail simulations, while the modified pyrcel model serves as a benchmark to highlight model discrepancies. After examining these discrepancies, we use the modified pyrcel

model to perform a suite of sensitivity analyses, studying the influence of the prescribed vPM size distribution characteristics and hygroscopicity, ambient temperature, and aircraft/fuel properties.

In Sect. 2, we provide an overview of elementary contrail thermodynamics (Sect. 2.1), the description of particle activation in contrail mixing plumes (Sect. 2.2) and implications for the SAC (Sect. 2.3). Next, we provide an overview of the underlying model framework (Sect. 3.1) and extensions (Sect 3.2 and Sect. 3.3), and the assumed particle characteristics used in this work (Sect. 3.4). We then evaluate the two models in Sect 4.1 and Sect 4.2, justifying use of the modified pyrcel model for several sensitivity analyses (Sect 4.3 – Sect. 4.6). Finally, we comment on several implications for future modelling studies and wider impacts (Sect. 5).

## 2. Background

Here, we review the thermodynamic pathway to contrail formation and define the SAC framework (Sect. 2.1). We describe several fundamental principles of particle activation (Sect. 2.2) and use these to justify an extension to the original SAC framework (Sect. 2.3). Finally, we provide a theoretical framework for the parcel models used within this work (Sect. 2.4).

### 2.1 Thermodynamics of contrail formation

A cruising aircraft releases combustion emissions at a typical engine exhaust temperature $T_E \sim 600$ K, which depends on the engine architecture and bypass ratio (Cumpsty, 2003). The emissions are dispersed into the turbulent jet plume of the aircraft and are rapidly diluted by upper tropospheric air, cooling to the ambient temperature ($T_A$) within 1 s. We can use the partial pressure of water in the mixing plume ($p_{v,M}$) to navigate the transition between exhaust and ambient conditions. For isobaric conditions, this property scales linearly between exhaust and ambient conditions (Rogers and Yau, 1996)

$$G = \frac{dp_{v,M}}{dT} = \frac{p_{v,E} - p_{v,A}}{T_E - T_A},$$ (1)

where $p_{v,E}$ and $p_{v,A}$ represent the partial pressure of water in the exhaust and ambient air, respectively, and $G$ is the gradient of the mixing line. Provided we know the ambient conditions ($p_{v,A}$, $T_A$) and $T_E$, to fully constrain Eq. (1) we must also define either $p_{v,E}$ or $G$. Conveniently, the latter has been shown to depend on the aircraft and fuel properties (Schumann, 1996),

$$G = \frac{c_p EI_w P_T}{\varepsilon Q (1 - \eta)},$$ (2)

where $P_T$ is the ambient air pressure, $c_p$ is the isobaric specific heat of air (1004 J kg$^{-1}$), $EI_w$ is the mass emission index of water vapour, $\varepsilon$ is the ratio of gas constants for water vapour and dry air (0.622), and $\eta$ is the overall efficiency such that (1 -

$\eta$) represents the fraction of the total heat released per mass of fuel burned ($Q$) that is transferred to the exhaust gas. We can describe $p_{v,M}$ by re-arranging Eq. (1),

$$p_{v,M} = p_{v,A} + G(T - T_A). \tag{3}$$

Often, it is useful to describe how the water or ice saturation ratio of the plume ($S_{v,M}$ and $S_{i,M}$, respectively) evolves during the mixing process (Fig. 1). Here, we define $S_{v,M}$ and $S_{i,M}$ within the mixing plume as follows,

$$S_{v,M} = \frac{p_{v,M}}{p_{liq}(T)}, \tag{4}$$

$$S_{i,M} = \frac{p_{v,M}}{p_{ice}(T)}. \tag{5}$$

$p_{liq}(T)$ and $p_{ice}(T)$ represent the saturation vapour pressure above a plane surface of supercooled liquid water or ice, respectively (Murphy and Koop, 2005). In turn, we use Eq. (4) to define the liquid water supersaturation within the contrail mixing plume as $s_{v,M} = S_{v,M} - 1$.

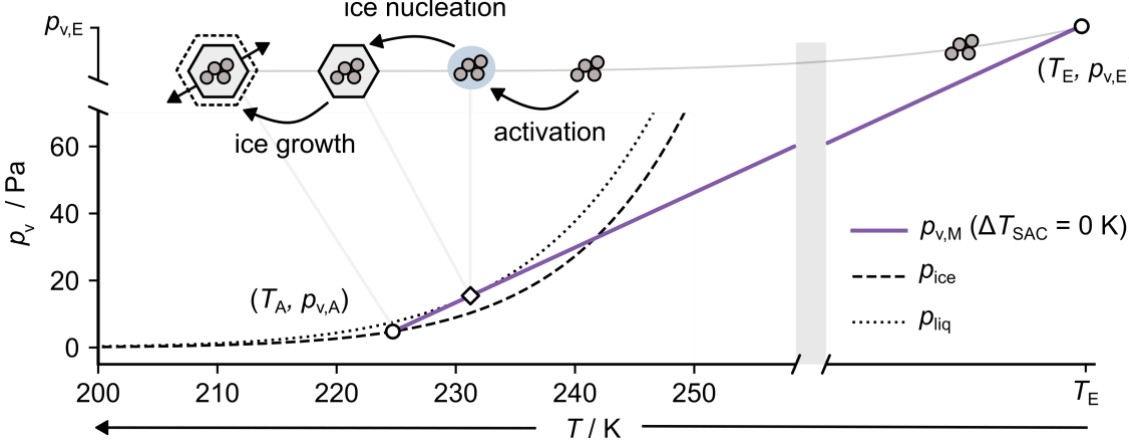

**Figure 1: Partial pressure of water vapour ($p_v$) and temperature of the plume ($T$) during contrail mixing (purple line) under threshold**
**conditions ($T_A = T_{SAC}$). Saturation vapour pressures above ice ($p_{ice}$) and water ($p_{liq}$) are also shown. Finally, a schematic representation of particle emission, activation and ice nucleation is shown. The latter two occur simultaneously within the original SAC framework. Here, we assume typical aircraft and fuel properties of ($P_T$, $EI_w$, $Q$, $\eta$) of 25 kPa, 1.23 kg·kg$^{-1}$, 43.2 MJ and 0.3 for temperatures $T_A$ and $T_E$ of ~ 224 K and 600 K respectively.**

To illustrate several features associated with contrail formation, we have shown an example of mixing behaviour for typical aircraft and fuel properties, see Fig. 1. The first feature is the crux of the SAC: contrail formation can only occur if $S_{v,M}$ > 1 ($s_{v,M}$ > 0). However, this represents only an approximate requirement for the activation of particles into liquid droplets, which we will discuss further in Sect. 2.2 and 2.3. Nevertheless, as originally defined, the SAC may be interpreted quantitatively as

$$\frac{dp_{liq}(T)}{dT}\bigg|_{T_{SAC}} = G, \tag{6}$$

where the Schmidt-Appleman threshold temperature ($T_{SAC}$) as the temperature at which the contrail mixing line tangentially intersects $p_{liq}$. Under this definition, $T_{SAC}$ represents the highest temperature at which contrail formation can occur in a water-saturated plume. The mixing behaviour presented in Fig. 1 is consistent with this threshold condition. Notably, for lower $T_A$ and fixed $p_{v,A}$, the mixing line would be translated towards cooler $T$ by an amount $\Delta T_{SAC}$ and the plume would become supersaturated during mixing. Hence, the parameter $\Delta T_{SAC}$ provides a useful indicator for the degree of supersaturation experienced by a contrail mixing plume. Importantly, the SAC implies that the activation of particles into liquid droplets occurs before ice nucleation takes place.

This water saturation constraint is consistent with observations of young contrails (Kärcher et al., 2015) and implies a formation pathway of condensation followed by homogeneous ice nucleation (Murray et al., 2012). As such, we omit heterogeneous ice nucleation in our analysis, which would otherwise contradict this mechanism. The water saturation constraint is also motivated by particle properties because: (a) aircraft-emitted nvPM particles have previously been shown as inactive heterogeneous ice nuclei (Testa et al., 2024); (b) only a small subset of ambient tropospheric aerosol particles (mineral dust, ammonium sulfate and amorphous organic) are efficient heterogeneous ice nuclei (Kärcher et al., 2023) and these have low number concentrations at cruise altitudes (relative to lean-burn $EI_{nvPM}$); and (c) plume vPM is likely to be at least partly soluble and therefore inactive as heterogeneous ice nuclei. Therefore, in the context of nvPM, the water saturation constraint implies that particles first activate to form water droplets, after which they act as passive inclusions, with rates of homogeneous ice nucleation controlled by the ambient temperature and volume of adsorbed water. In the context of plume vPM and ambient particles, we extend this analysis and demonstrate that the process of ice formation is again strictly sequential, with particle activation occurring before homogeneous ice nucleation, see SI S1. For this reason, contrail formation is dictated by the conditions under which particle activation takes place, which are addressed in Sect. 2.2.

## 2.2 Aerosol particle activation and the κ-Köhler framework

The equilibrium saturation ratio over a particle may be described as a balance of two interactions (Seinfeld and Pandis, 1998). The first of these is the Kelvin (also commonly referred to as the "surface curvature") effect. Consider the situation whereby a particle with (dry particle) diameter $d_d$ is activated to form a pure liquid droplet with a (wet particle) diameter of $d_w$. Due to surface tension at the water-air interface, a decrease in the droplet diameter leads to an increase in the partial pressure of water vapour at the droplet surface, and hence an increase in the water saturation ratio ($S_v$). This effect is parameterized as

$$S_v = a_w e^{\left(\frac{4\sigma_S M_w}{RT\rho_w d_w}\right)}, \tag{7}$$

where $\sigma_s$ is the surface tension of the liquid/air interface, $M_w$ is the molar mass of water, $\rho_w$ is the density of the liquid, $R$ is the global gas constant, $T$ is the temperature and $a_w$ is the activity of water in solution (Petters and Kreidenweis, 2007). For particles that are entirely insoluble, the liquid surrounding the particle will be pure water and we can approximate $a_w = 1$, reducing Eq. (7) to the Kelvin equation (Pruppacher and Klett, 1980). This simplified treatment of insoluble particle activation has been revisited by several studies that consider multilayer adsorption mechanisms (Kumar et al., 2009; Sorjamaa and Laaksonen, 2007), although the application of these approaches is limited by the availability of experimental data. In contrast, if the particle is not entirely insoluble (for example, the insoluble core may contain a soluble coating), the value of $a_w$ (and $S_v$) is reduced, thus defining the second, solute effect. In this case, the activity of water in solution may be treated using the parameterization proposed by Petters and Kreidenweis (Petters and Kreidenweis, 2007). Here, the particle is described using a hygroscopicity parameter ($\kappa$) which is related to the bulk material solubility (Sullivan et al., 2009), and $a_w$ is given by

$$a_w = \frac{d_w^3 - d_d^3}{d_w^3 - d_d^3(1-\kappa)}, \tag{8}$$

where, for an internally mixed particle, $\kappa$ is given as a volume-weighted sum over the individual components. Notice that Eq. (7) collapses to $a_w = 1$ if $\kappa = 0$, which is the approximate limit for a non-hygroscopic (insoluble) particle. Above this limit, $\kappa$ is typically represented on a logarithmic scale, with slightly hygroscopic materials ($\kappa < 0.01$) less capable of water uptake than very hygroscopic materials ($\kappa > 0.5$) (Petters and Kreidenweis, 2007). Combining Eq. (7) and Eq. (8), we recover the final form of $\kappa$-Köhler theory,

$$S_v = \frac{d_w^3 - d_d^3}{d_w^3 - d_d^3(1-\kappa)} e^{\left(\frac{4\sigma_S M_w}{RT\rho_w d_w}\right)}. \tag{9}$$

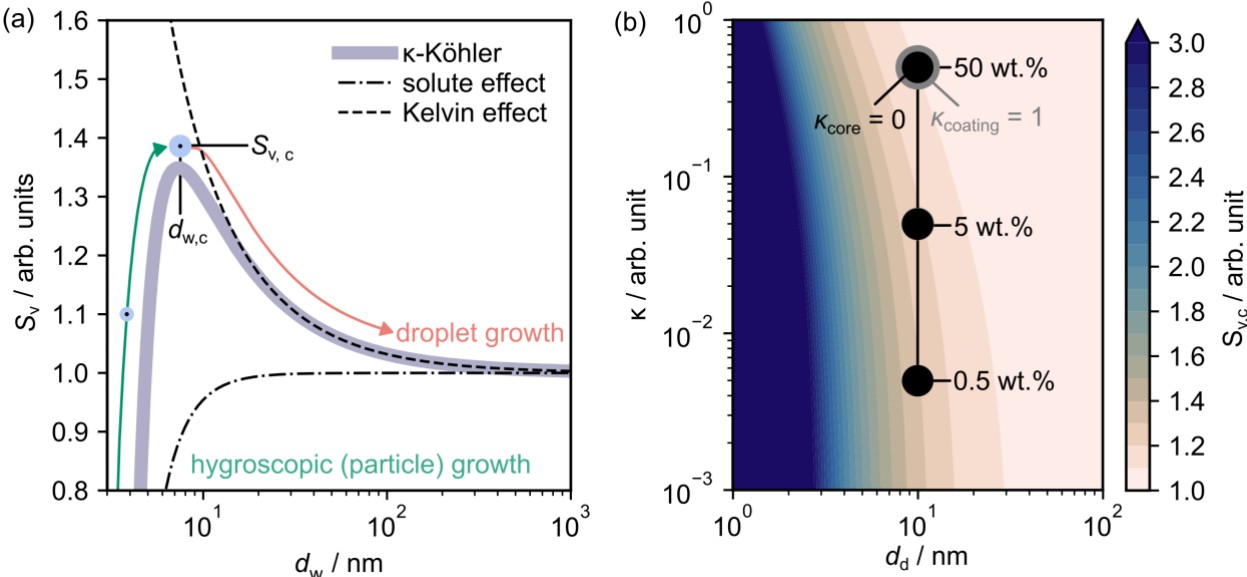

Figure 2: (a) Water saturation ratio ($S_v$) with increasing wet particle diameter ($d_w$) as the combination of the solute and Kelvin effects. The critical diameter ($d_{w,c}$) and the associated critical saturation ratio ($S_{v,c}$) mark the point at which a particle is considered activated (b) $S_{v,c}$ is shown as a function of the dry particle diameter ($d_d$) and particle hygroscopicity. Values for insoluble ($\kappa = 0$) particles with variable highly soluble ($\kappa = 1$) coatings have been shown for reference.

According to Eq. (9), $S_v$ increases with $d_w$ up to a maximum at ($d_{w,c}, S_{v,c}$), beyond which it declines, see Fig. 2a. These critical parameters may be estimated for a given particle type and at a given $T$ by setting the derivative of Eq. (9) equal to zero and solving numerically. In the context of particle activation, it is more instructive to explore how $d_w$ depends on the ambient saturation ratio, remaining mindful that the inverse function $d_w = d_w(S_v)$ is multivalued. To that end, as the ambient $S_v$ is increased over a particle, initially growth is regulated by the solute effect and the particle diameter increases by slow, hygroscopic growth. Upon further increasing $S_v$, the relative magnitudes of the Kelvin and solute effects balance at the critical point ($S_{v,c}, d_{w,c}$) (Seinfeld and Pandis, 1998). Here, the particle is said to have been activated, and subsequent growth is regulated by kinetic droplet growth outlined in SI S2. In summary, a particle is considered activated to form a water droplet once the ambient $S_v$ exceeds $S_{v,c}$.

Figure 2b shows that $S_{v,c}$ exhibits three key trends: (i) it increases with both $d_d$ and $\kappa$; (ii) it is more sensitive to changes in $d_d$ than $\kappa$ (i.e., $\frac{dS_{v,c}}{dd_d} \gg \frac{dS_{v,c}}{d\kappa}$); and (iii) it approaches 1 in the limit ($\kappa, d_p$) → (1, ∞), $S_{v,c}$ → 1, for which additional justification is provided in SI S3. In Figure 2b we have also shown how $S_{v,c}$ varies for a series of mixed particles with insoluble ($\kappa = 0$) cores and variable coatings of hygroscopic ($\kappa = 1$) material (0.5 wt.% - 50 wt.%) with similar total dry particle diameters $d_d = 10$ nm. These highlight the importance of soluble coatings when considering plume particle activation, which is a feature we will return to in Sect. 4.5.

## 2.3 Extended Schmidt-Appleman framework

As discussed in Sect. 2.1, the SAC assumes that contrails are formed if the mixing line becomes (super)saturated with respect to water during plume evolution ($S_v \geq 1$). However, as outlined in SI S3, this criterion cannot be reconciled with the minimum requirement for (even hygroscopic) particle activation.

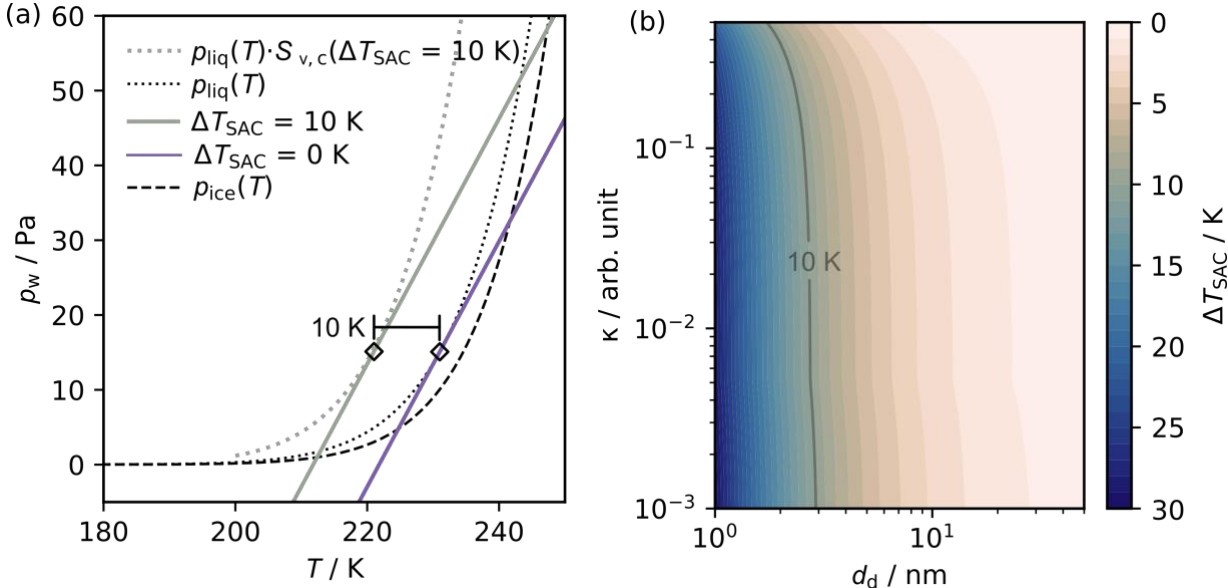

Figure 3: (a) Saturation vapour pressure above water ($p_{liq}$) (purple, dotted) and a contrail mixing line (purple, solid) for threshold conditions, $\Delta T_{SAC} = 0$ K. Particle properties ($d_d$, $\kappa$) have been chosen so that the modified saturation vapour pressure above water $p_{liq} \cdot S_{v,c}$ ($d_d$, $\kappa$) (grey/green, dotted) translates threshold conditions by 10 K, so $\Delta T_{SAC} = 10$ K (grey/green, solid). (b) Extension of panel (a) to show dependence of $\Delta T_{SAC}$ on $d_d$ and $\kappa$. Combinations of particle properties resulting in $\Delta T_{SAC} = 10$ K have been shown (grey/green, solid). In all cases, $S_{v,c}$ is estimated conservatively using $T = 220$ K.

Instead, suppose that a contrail mixing contains a single particle mode with properties ($d_d$, $\kappa$). Now, the criterion for contrail formation is that the contrail mixing line must contact $p_{liq} \cdot S_{v,c}(T, d_d, \kappa)$ at least once during the mixing process. Figure 3a shows that when these particle properties are included, the threshold requirement for activation (previously water saturated conditions, black dotted line) is elevated (grey/green dotted line). Using Eq. (9), the particle dependence of $\Delta T_{SAC} = \Delta T_{SAC}(T, d_d, \kappa)$, can be determined and is shown in Fig. 3b. Alternatively, we can consider the parameter $\Delta T_{SAC}(T, d_d, \kappa)$ presented in Fig. 3b as an *effective* Schmidt-Appleman temperature, $T_{SAC}(d_d, \kappa)$, the temperature below which contrail formation takes place when the plume contains particles with properties $d_d$ and $\kappa$. Figure 3b also enables us to estimate for which particle types the SAC framework (as outlined in Sect. 2.1) remains applicable, under a specified temperature uncertainty.

For particles with $\kappa > 0.6$ and/or $d_d > 20$ nm, $\Delta T_{SAC}(T, d_d, \kappa) < 2$ K, which is close to the typical uncertainty associated with low-altitude temperature measurements during *in-situ* campaigns, $\Delta T = \pm 1.9$ K (Bräuer et al., 2021). Therefore, for "soot-rich" emissions, where the typical nvPM particles have $d_d > 20$ nm, the condition for applying the original SAC is valid within measurement uncertainty. This explains the success of this approach over the past decades and across various measurement campaigns (Schumann et al., 2017). However, with reduced soot emissions, exhaust material will be enriched with smaller plume vPM particles. Under these circumstances, the SAC is less effective at predicting the temperature at which *most particles* activate, see Fig. 3b.

## 3. Model

Here, we outline the fundamental relations used to describe a parcel model for contrail formation (Sect. 3.1), including a basic overview of the original K15 model. Next, we outline several extensions made to the K15 model, so that it can be applied to treat plume vPM activation (Sect. 3.2). Then, we outline modifications made to a preexisting cloud parcel model, pyrcel (Rothenberg and Wang, 2016), to simulate the early stages of contrail formation (Sect. 3.3). Finally, we describe the parameter space used in our model simulations (Sect. 3.4).

### 3.1 Contrail parcel model

As discussed in Sect. 2.2, the critical parameter governing uptake of water onto particles is the water saturation ratio, $S_v$. The water saturation ratio within a parcel of air can be described as

$$S_v = \frac{P_T}{p_{liq}(\varepsilon + w_v)} w_v, \tag{10}$$

where $\varepsilon$ (= 0.622) is the ratio of the molar mass of water (18.02 gmol$^{-1}$) to the molar mass of dry air (28.97 gmol$^{-1}$) and $w_v$ is the water vapour mixing ratio, the mass of water vapour per unit mass of dry air contained within the parcel. We can simplify Eq. 10 further using several features of contrail mixing. Within a contrail, the maximum partial pressure of water vapour in the plume is given by the maximum of $S_v \, p_{liq}$. This is bounded by $S_{v,M} \, p_{liq}$, since isobaric mixing is the only process that acts to increase the parcel saturation ratio. Assuming that the ambient environment is ice-*saturated*, we also know that $p_{v,M}$ is bounded by $(p_{v,M})_{max} = p_{ice}(T_{SAC}) + G(T_E - T_{SAC})$. Rearranging Eq. (10), we can therefore bound $w_v$ as

$$w_v < \frac{\varepsilon (p_{v,M})_{max}}{P_T - (p_{v,M})_{max}}. \tag{11}$$

Using $T_E = 600$ K, $T_{SAC} = 224$ K, $G = 1.64$ PaK$^{-1}$ and P$_T = 23000$ Pa, we have that $w_v < 0.02$. Therefore, the inequality $w_v \ll \varepsilon$ is satisfied for all conditions, which enables us to simplify Eq. (10) as

$$S_v = \frac{P_T}{p_{liq}\varepsilon} w_v. \tag{12}$$

Next, we assume that contrail mixing occurs at constant atmospheric pressure so that upon differentiating Eq. (12) with respect to time and collecting like-terms, we have

$$\frac{dS_v}{dt} = \frac{P_T}{p_{liq}\varepsilon} \frac{dw_v}{dt} - S_v \frac{1}{p_{liq}} \frac{dp_{liq}}{dt}, \tag{13}$$

$$\frac{dw_v}{dt} = -\left(\frac{dw_c}{dt}\right)_{microphysical} + \left(\frac{dw_{v,M}}{dt}\right)_{mixing}. \tag{14}$$

Equation (14) is a statement of mass conservation: any change in the water vapour content of the parcel must result from either particle microphysical processes (i.e., particle activation and droplet/ice crystal growth) or from plume mixing, provided there is no entrainment of ambient water vapour. Using Eq. (10) we can express the mixing term as

$$\left(\frac{dw_{v,M}}{dt}\right)_{mixing} = \dot{T} \frac{d}{dT}\left(\frac{p_{v,M}\varepsilon}{P_T}\right) = \frac{\varepsilon}{P_T} G\dot{T}, \tag{15}$$

where $\dot{T}$ is the cooling rate in the plume. Then, combining Eq. (12) – (15), we arrive at the governing equation

$$\frac{dS_v}{dt} = \frac{1}{p_{liq}} G\dot{T} - S_v \frac{1}{p_{liq}} \frac{dp_{liq}}{dt} - \frac{P_T}{p_{liq}\varepsilon} \frac{dw_c}{dt}. \tag{16}$$

This is almost identical to the result given by (Kärcher et al., 2015). However, we find that on the right-hand side of Eq. (16), the actual plume saturation ratio ($S_v$) is present rather than the saturation ratio assumed from mixing alone ($S_{v,M}$). This equation can be solved numerically, for example by using a numerical parcel model. However, the equation cannot be solved analytically. Under these circumstances, a solution may be derived by first approximating $S_v \sim S_{v,M}$ on the right-hand side of Eq. (16) as in the original K15 model, which effectively decouples the equation. In this purely analytical case, we can
generalize Eq. (16) in the form

$$\frac{dS_v}{dt} = P_w - L_w, \tag{17}$$

where $P_w$ represents the rate of change of the water saturation ratio without particles in the plume, and $L_w$ represents the rate of change of the water saturation ratio resulting from (a) particle activation to form water droplets and (b) growth of droplets and/or ice crystals. The description for $P_w$ follows from the description of contrail mixing outlined in Sect. 2.1 and is given by (Kärcher et al., 2015)

$$P_w = \frac{1}{p_{\text{liq}}} G\dot{T} - S_{v,M} \frac{1}{p_{\text{liq}}} \frac{dp_{\text{liq}}}{dt} = \frac{dS_{v,M}}{dt}, \tag{18}$$

Using the description of mixing provided by (Kärcher et al., 2015), next we will outline the equations necessary to estimate the cooling rate. For a full description of the parameters and underlying approximations, the reader is directed to the original text and references therein. The contrail mixing plume cools as it is diluted with ambient air according to the dilution factor

$$D(T) = \frac{T - T_A}{T_E - T_A}, \tag{19}$$

which describes how the cross-sectional area of the plume decreases with decreasing ambient temperature. The temporal evolution of the plume is then described by

$$\frac{dT}{dt} = -\beta \frac{T_E - T_A}{\tau_m} D^{1+1/\beta}, \tag{20}$$

$$t = \tau_m D^{-1/\beta}, \tag{21}$$

where $\tau_m$ is the timescale over which the contrail mixing parcel is unaffected by entrainment and $\beta$ is a constant dilution parameter. Note that we choose to omit effects associated with latent heat within our simulations, which we found to have a negligible impact on projected ice crystal number concentrations. Therefore, if we have knowledge of the aircraft and ambient conditions, using Eq. (19) – (21), we are now able to describe $P_w$ within the contrail mixing parcel at a given temperature. These relations also govern particle dilution within the cooling contrail mixing parcel. To that end, the total particle number concentration of aircraft ($n_T$) and ambient ($n_{a,T}$) particle modes are respectively

$$n_T = \frac{D(T)\rho_a(T)}{N_0} \text{EI}, \tag{22}$$

$$n_{\mathrm{a},T} = \frac{T_A}{T}[1 - D(T)]n_{\mathrm{a},0}, \tag{23}$$

where $\rho_a$ is the mass density of ambient air, EI represents the emission index of the aircraft mode, $n_{\mathrm{a},0}$ represents the ambient number concentration far from the plume and $\mathcal{N}_0$ is the mass-based mixing ratio according to the K15 model. Particle types are assumed to conform to lognormal particle size distributions and described using a fixed (total) hygroscopicity parameter, $\kappa$. For this reason, only a fraction of the total particles within a given mode will be able to activate under a given set of ambient conditions. The activated number concentration can be estimated by integrating over the particle size distribution using Eq. (9), using a methodology outlined in SI S4. The second term in Eq. (17), $L_{\mathrm{w}}$, may be expressed as (Korolev and Mazin, 2003)

$$L_{\mathrm{w}} = \frac{P_T}{e_s^0 \varepsilon} \frac{dw_c}{dt}; \quad \frac{dw_c}{dt} = \frac{4\pi \rho_w}{\rho_a} \sum_{i=1}^{n} n_{w,i} r_{w,i}^2 \dot{r}_{w,i}, \tag{24}$$

where $n_{w,i}$ is the number concentration of droplets, $r_{w,i}$ is the droplet radius and $\dot{r}_{w,i}$ is the droplet growth rate of these droplets, see SI S2. In Eq. (24), we have summed over contributions from each of the droplets at a given time $t$. Another useful formulation of Eq. (24) is

$$L_w = \frac{P_T}{p_{\mathrm{liq}} \varepsilon} \frac{4\pi \rho_{\mathrm{w}}}{\rho_{\mathrm{a}}} \int_0^\infty r_0 \frac{dn_{\mathrm{w}}}{dr_0} \int_{-\infty}^{t} \dot{n}_{\mathrm{w}}(t_0)\, r_{\mathrm{w}}^2(t, t_0) \dot{r}_{\mathrm{w}}(t, t_0)\, dt_0, \tag{25}$$

where $\dot{n}_{\mathrm{w}}(t_0)\, dt_0$ is the number density of aerosol particles that activate between times $t_0$ and $t + dt_0$, $4\pi r_{\mathrm{w}}^2(\mathrm{t}, t_0)/\upsilon$ is the volumetric flux of water towards water droplets at time $t$, that first formed at time $t_0 < t$, and $\dot{r}_{\mathrm{w}}(t, t_0)$ is the growth rate of these droplets at time $t$ (Kärcher and Lohmann, 2002a). We have also integrated over the size distribution of activated particles, $\frac{dn_{\mathrm{w}}}{dr_0}$. In Eq. (25), rather than summing over contributions from each droplet at a given time $t$, we have instead integrated over each time $t_0 < t$ and estimated the projected contribution towards the final time $t$. Equations (24) and (25) are therefore equivalent representations, that are suited to different applications.

Finally, the contrail mixing behaviour can be described using $P_{\mathrm{w}}$ in combination with either expression for $L_w$. The formulation in Eq. (24) is used in traditional cloud parcel models using discretized particle and droplet size distributions, which enables Eq. (17) to be solved iteratively as a function of time. This forms the basis for the pyrcel model, which is described in more detail in Sect. 3.3. On the other hand, in the K15 model, the formulation in Eq. (25) is solved analytically at each timestep, assuming that particle activation and droplet growth at time $t < t_0$ does not affect $S_v$ at time $t_0$. At each time $t$, we estimate the critical droplet number concentration required to prevent any further change in the plume saturation ratio $\left(\frac{dS_v}{dt} = 0\right)$. Using Eq. (25), we can define the mean supersaturation loss rate per droplet as $R_{\mathrm{w}} = L_{\mathrm{w}}/n_{\mathrm{w}}(t)$ and reformulate this critical droplet number concentration as $n_{\mathrm{w}}^{(2)}(t) = \frac{P_{\mathrm{w}}}{R_{\mathrm{w}}}$. As time evolves and $S_v$ increases, so too will the number concentration of droplets,

$n_w^{(1)}(t)$. At some threshold time, the number concentration of droplets will be equal to the critical droplet number concentration i.e., $n_w^{(1)}(t) = n_w^{(2)}(t)$. Assuming the plume saturation ratio has equilibrated, no further activation can take place after this time (Twomey, 1959). Hence, this condition defines a lower bound for the maximum number of water droplets that can form during the mixing process, providing an estimate for the contrail ice crystal number concentration. This process forms the basis of the K15 model approach, for which several extensions are described in Sect. 3.3.

## 3.2 Model extensions for K15

The original K15 model only considers aircraft-emitted nvPM and ambient particles, both of which contain a single particle type and are described using assumed lognormal particle size distribution characteristics and $\kappa$. This approach is applicable to nvPM particles because their size distribution characteristics do not change significantly over time and $\kappa$ is dominated by the insoluble nvPM fraction. This approach is also applicable to ambient solution droplets, because once entrained into the plume, these droplets are readily activated due to the high ($S_v \gg 1$) plume supersaturations. Therefore, small changes in their size distribution characteristics that are not captured by the K15 model are relatively insignificant. Considering this, our first extension to the K15 model is to introduce plume vPM (Yu et al., 1998) as an additional particle type. We define plume vPM to consist of condensable gaseous material including (but not limited to) sulphuric acid, organic material, and engine lubrication oil. Simulations show that the size distribution characteristics of plume vPM evolve over time due to condensational growth and coagulation (Kärcher et al., 2000). However, here we are concerned with the contrail formation phase $0.1 - 1$ s into plume dilution, so we prescribe plume vPM properties expected at these timescales. As such, our model cannot be used to *predict* the properties of plume vPM. Instead, we *prescribe* plume vPM using a fixed emission index, $\kappa$ and particle size distribution characteristics, and evaluate the model sensitivity to these variables under typical ranges. In SI S5 we also investigate the use of a more complex ambient particle description comprising 7 independent particle types. The particle size distribution characteristics and hygroscopicity of these particle types are estimated using outputs from the global aerosol-climate model ECHAM-HAM (Zhang et al., 2012). Overall, we suggest that incorporating these additional ambient particle types could enhance $AEI_{ice}$ by an order of magnitude when most ice crystals form via ambient particles. Next, there are several microphysical processes within the K15 model which are described using analytical approximations. Our extensions focus on either improving the accuracy of these approximations or removing them altogether in favour of numerical solutions. Firstly, the analytical approximation used to estimate the activated number concentrations for a given particle type can lead to overpredictions for $S_v < S_{v,c}$ (GMD, $\kappa$, $T$) (SI S6). To address this, we updated the methodology to estimate activated particle number concentrations using an exact analytical solution.

Secondly, the K15 model estimates $S_{v,c}(T, d_d, \kappa)$ using an approximate analytical solution derived from Eq. (9). We compared this with the full numerical solution and show that the K15 approximation systematically underpredicts critical supersaturations for $\kappa$ ($< 0.2$), with larger discrepancies upon reducing GMD (SI S7). Hence, we solve Eq. (9) numerically when identifying critical supersaturations, in line with (Bier and Burkhardt, 2022).

395   The aforementioned approximations each indirectly relate to the $P_w$ term, however the final approximation relates to the $L_w$ term. The methodology for estimating contrail ice crystal number concentrations outlined in K15 relies on analytical solutions for both $P_w$ and $L_w$. In SI S8 we overview the relevant equations used to estimate $L_w$ and give an account of the various approximations used in its derivation. After considering these, we suggest the following modification. The solution to $L_w$ provided in K15 is valid only for when particle activation occurs more quickly than droplet growth i.e., the "slow-growth"

400 regime (Kärcher and Lohmann, 2002b). After incorporating a plume vPM particle mode, this condition is not necessarily satisfied, so we use a regime-independent analytical solution (Kärcher et al., 2006).

   There are two outstanding limitations in the extended K15 model that cannot be resolved. First, we found no means of introducing the Kelvin correction into the solution for $L_w$ (via the droplet growth equation). Omitting the Kelvin effect in the droplet growth equation acts to increase the radial growth rate (SI S2). Hence, the ability for droplets to deplete the plume

405 supersaturation is increased and so a lower droplet number concentration is required to quench the ambient supersaturation. This effect is relevant for smaller particles such as plume vPM, where the magnitude of the Kelvin correction is increased. Second, an analytical solution for the inner integral of Eq. (25) requires that $L_w$ is independent of the plume supersaturation, so that droplet formation at time $t_0 < t$ does not deplete the plume supersaturation at time $t$. However, as defined in Eq. (25), $L_w$ is coupled to the plume supersaturation via the droplet growth equation. We anticipate this may be problematic when two

410 or more particle types with comparable number concentrations but different $S_{v,c}$ coexist. Here, one particle type will preferentially activate and grow, subsequently depleting the supersaturation available for the second particle type and thereby inhibiting its activation and growth. For this reason, the extended K15 does not account for the temporal depletion of supersaturation or competition between particle types.

## 3.3 Model modifications for Pyrcel

415 Pyrcel is a 0D cloud parcel model used to simulate the evolution of a parcel of air as it ascends adiabatically at a constant updraft speed (Rothenberg and Wang, 2016). The model is initialized by defining the aerosol population using lognormal particle size distributions, which are discretized into bins. These particle properties are used in combination with the initial ambient conditions to establish an equilibrium wet particle size distribution. Next, the conservation relations Eq. (11, 13, 21) are iterated forward in time and the wet particle radii within each bin are tracked using a Lagrangian grid. For a fuller

420 description of the underlying model mechanics, the reader is directed to the original paper and references therein (Rothenberg and Wang, 2016). In contrast to the K15 model, pyrcel explicitly considers the feedback of water vapour between different droplet populations and $P_w$. However, pyrcel was originally written to describe the formation of warm-phase clouds, therefore several modifications were made for application to contrail formation.

   Firstly, we replace the original $P_w$ term (deriving from parcel updraft) with the definition in Eq. (18). Secondly, we

425 remove the time-dependence of altitude and pressure, under the assumption of an adiabatic expansion at constant atmospheric pressure as described in Sect. 3.1. Finally, we derive initial particle number concentrations for the aircraft modes from the associated emission indices using Eq. (18, 19). After these changes, the modified pyrcel model and extended K15 models

provide an identical description of $P_w$. However, the coupling of $P_w$ to $L_w$ is fully represented only within the modified pyrcel model.

Additionally, we also extended pyrcel to include homogeneous ice nucleation within growing water droplets. For a complete description of the relations used to arrive at an estimate for the homogeneous ice nucleation temperature, the reader is directed to (SI S1). In brief, we estimate the rate of homogeneous ice nucleation within a droplet using its water activity (Koop et al., 2000), identified using Eq. (8) as in (Lewellen, 2020). The probability for ice nucleation within the droplet is then determined by assuming a pulse-like freezing process according to the methodology employed in K15. If the probability equals

unity, we assume that the supercooled droplet freezes. The growth of ice crystals by deposition of (ice) supersaturated water vapour is then described using equations analogous to those in SI S2, where water saturation in the numerator is replaced with ice saturation, which represents the driving force for vapour deposition. For additional information on the exact form of these expressions, the reader is directed towards (Pruppacher and Klett, 1980). As outlined above, our implementation is sufficiently flexible to capture homogeneous ice nucleation events within droplets that may not (yet) be considered activated ($d_w < d_{w,c}$).

### 3.4 Assumed particle properties

We use a conservative parameter space for the properties of each particle mode, see Table 1. We note that both $EI_{nvPM}$ and the particle size distribution characteristics can change with use of sustainable aviation fuel (SAF) (Märkl et al., 2024; Schripp et al., 2022), however these fall within larger uncertainties in our simulations. Recently, contrail parcel model simulations have been performed assuming a smaller $GMD_{nvPM}$, consistent with the primary particle diameter rather than the aggregate particle

diameter (Yu et al., 2024). While the ice-nucleating ability of larger soot particles has been shown to depend on particle morphology (Marcolli et al., 2021), there is a paucity of data available for activation and ice nucleation on surrogate aircraft nvPM. Therefore, here we choose to identify $GMD_{nvPM}$ as the aggregate particle diameter in line with other studies (Kärcher et al., 2015). Nevertheless, we evaluate the model sensitivity to the choice of nvPM primary particle or aggregate diameter in detail (see SI S10) and find that (i) $AEI_{ice}$ predictions agree within 40% and (ii) the functional behaviour is similar. Finally, we

assume that the nvPM mode contains a 1% wt. coating of soluble material as observed for ground-based emissions testing at moderate fuel sulphur content (FSC) (Gysel et al., 2003), and is therefore characterised by $\kappa = 0.005$ in accordance with the original K15.

    The properties of the ambient mode are taken directly from the original K15 model inputs. We argue that these properties are sufficient to investigate the competition between the three modes under most conditions, however a more

sophisticated description would be required to examine sensitivity if most ice crystals formed via ambient particles (Bier et al., 2023), see SI S5.

    For the plume vPM mode, we identify a provisional size range of 1 nm < $GMD_{vPM}$ / nm < 4 and $GSD_{vPM}$ of 1.3 (Kärcher et al., 2000; Yu et al., 1998) Similarly, we assume a fixed emission index $EI_{vPM} = 10^{17}$ kg$^{-1}$ in line with modelling and observational evidence (Arnold et al., 2000; Haverkamp et al., 2004; Schumann et al., 2002; Sorokin and Mirabel, 2001).

This is also supported by newer observational evidence (Voigt et al., 2025) that the total particle emission index (particle sizes

> 5 nm) is on the order $10^{15}$ kg$^{-1}$, which is consistent with EI$_{vPM}$ = $10^{17}$ kg$^{-1}$ if GMD$_{vPM}$ = 2.7 nm and GMD$_{vPM}$ = 1.3. Concerning particle hygroscopicity, we assume a mixed plume vPM mode comprising sulphuric acid ($\kappa$ = 0.5) and condensable POM, for which we use a conservative lower bound of $\kappa$ = 0 in line with measurements of non-hygroscopic lubrication oil droplets (Ponsonby et al., 2024). However, we acknowledge that the value of $\kappa$ for condensable POM is likely to vary with changes in fuel and/or engine architecture. To that end, we evaluate our sensitivity analyses in the extreme cases of both sulphur-rich plume vPM (volume mixing ratio of 95:5 sulphuric acid: condensable POM, $0.05 \cdot \kappa_{POM} + 0.95 \cdot \kappa_{H2SO4} \sim 0.5$) and organic-rich plume vPM (volume mixing ratio of 5:95 for sulphuric acid: condensable POM, $0.95 \cdot \kappa_{POM} + 0.05 \cdot \kappa_{H2SO4} \sim 0.01$). In SI S9 we investigate the impact of assuming all plume vPM particles have the same chemical composition ($\kappa$). To achieve this, we divide the plume vPM mode into two distinct plume vPM modes, prescribing these with identical physical characteristics but dissimilar $\kappa$. We find that the results of the two-mode system are similar to that of the single-mode system with a modified value of $\kappa$. Therefore, we argue that our range of $\kappa$ is sufficient to capture a degree of variation in chemical composition *between* vPM particles.

In the above, we have described vPM characteristics at the timescale of particle activation and droplet growth in the nascent contrail mixing plume (~ 1 s). Separately, we can also estimate the (maximum) equilibrium plume vPM diameters ($d_{vPM}$) that would be reached at t >> 1 using a mass balance. Among other factors, the characteristics of the plume vPM particle size distribution are constrained by the quantity of available sulphuric acid and POM. The former contribution is produced by oxidation of FSC,

$$\mathrm{EI}_{H_2SO_4} = \frac{98}{32} \varepsilon_s \mathrm{FSC}, \tag{26}$$

where $\varepsilon_s$ is the conversion efficiency of sulphur to sulphuric acid, which is typically on the order of 3% (Yu et al., 2024). The latter contribution represents the sum of condensable gaseous emissions, which has previously been estimated in the range 1 – 40 mg/kg for relevant, combustion-related organics (Yu et al., 1999) in addition to potential non-combustion lubrication oil emissions of 110 mg/kg (Decker et al., 2024). As outlined in Sect. 1, we can assume that the plume vPM mode can be treated as monomodal, as the smaller mode (i) requires a higher critical supersaturation for activation and (ii) is effectively scavenged by the larger mode. Conservation of mass enables us to estimate $d_{vPM}$, for the plume vPM mode (Kärcher et al., 2000)

$$\frac{\pi}{6} d_{vPM}{}^3 \rho_{vPM} \mathrm{EI}_{vPM} = \frac{98}{32} \varepsilon_s \mathrm{FSC} + \mathrm{EI}_{POM}, \tag{27}$$

where $\rho_{vPM}$ is the density of the volatile mode. Using Eq. (27), we can therefore estimate $d_{vPM}$ for given EI$_{POM}$ and FSC, which will be explored in Sect. 4.5.

**Table 1: Assumed characteristics and associated ranges for each particle type used within our parcel model simulations. List of references in table: [1] (Arnold et al., 2000) [2] (Haverkamp et al., 2004) [3] (Schumann et al., 2002) [4] (Sorokin and Mirabel, 2001) [5] (Kärcher et al., 2000) [6] (Yu et al., 1998) [7] (Ponsonby et al., 2024) [8] (ICAO, 2024) [9] (Durdina et al., 2024) [10] (Petzold et al., 2005) [11] (Kärcher et al., 2015)**

| Mode properties | EI / kg$^{-1}$ | GMD / nm | GSD / no units | $\kappa$ / no units |
|---|---|---|---|---|
| Plume vPM | $10^{17}$ $[1-4]$ | $1-4$ [5] | 1.3 [6] | $0-0.5$ [7] |
| nvPM | $10^{12} - 10^{16}$ [8] | 35 [9] | 2.0 [9] | 0.005 [10] |
| ambient [11] | 600 cm$^{-3}$ | 30 | 2.2 | 0.5 |

## 4. Results

Here, we present the results of both the extended K15 model and modified pyrcel models. Based on these results, we then use the modified pyrcel model to run a set of sensitivity analyses and explore how the apparent emission index of ice crystals (AEI$_{ice}$) within a contrail depends upon particle properties (Sect. 4.3 and 4.5), $T_A$ (Sect. 4.4) and aircraft parameters (Sect. 4.6).

### 4.1 Contrail mixing behaviour

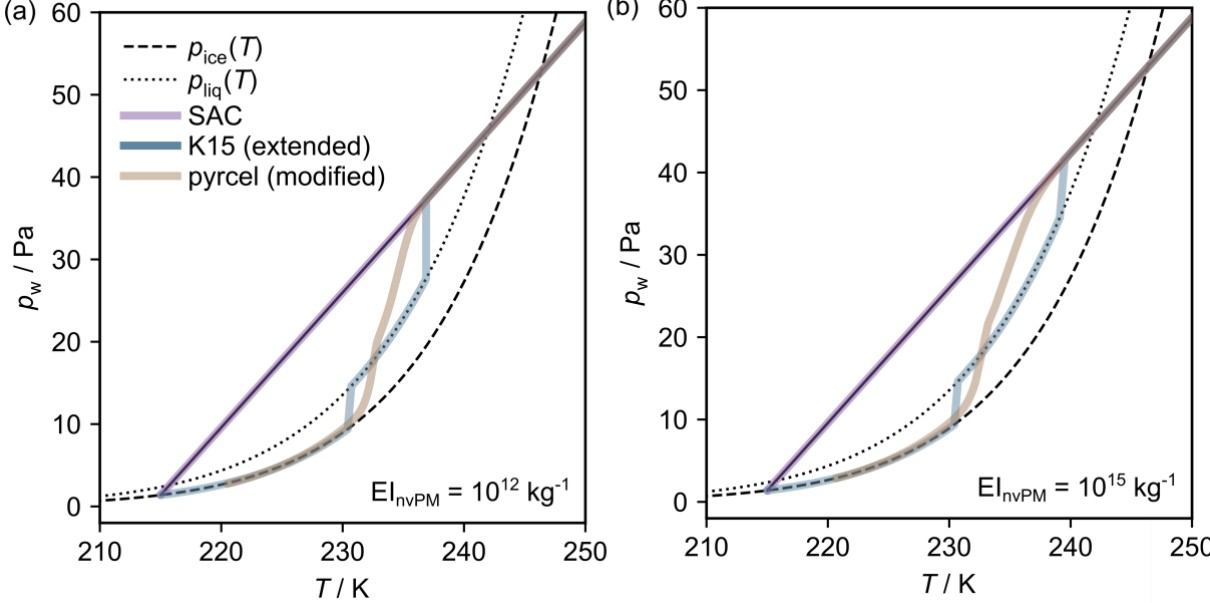

**Figure 4: Contrail mixing lines simulated with assumed EI$_{nvPM}$ of (a) $10^{12}$ kg$^{-1}$ and (b) $10^{15}$ kg$^{-1}$. For both scenarios, the mixing lines are simulated using the original SAC framework (purple lines), the extended K15 model (blue lines), and the modified pyrcel model (brown lines). Each of the simulations has been set to terminate at ice saturated conditions. These model predictions have been made with plume vPM (EI$_{vPM}$ = $10^{17}$ kg$^{-1}$).**

Figure 4 shows the typical mixing behaviour as predicted by extended K15 and modified pyrcel models. These are compared to profiles in the absence of any particle intervention (SAC-type mixing behaviour), which is encapsulated by $P_w$. In both the extended K15 model and modified pyrcel models, the partial pressure of water within the plume follows a similar trend. When the conditions in the plume are subsaturated (i.e., $S_v < 1$), both models align with SAC-type mixing behaviour since very few particles can activate. Hence, $L_w \sim 0$ and the mixing is dominated by $P_w$. However, once the plume becomes supersaturated, a proportion of the particles activate to form aqueous solution/water droplets. Here, $L_w$ gradually increases to a maximum, before equilibrating with $P_w$ at water saturated conditions.

This process of "activation-relaxation" (see warmer-temperature vertical regions of blue line in Fig. 4) is approximated within the extended K15 model by estimating the supersaturation at which this takes place and an associated timescale, see SI S8. Notice that because particles require supersaturated conditions to activate ($S_{v,c} > 1$), the onset temperature for "activation-relaxation" within the extended K15 model occurs after the point of intersection between the contrail mixing line and $p_{liq}$. These findings are in qualitative agreement with the modified pyrcel outputs, although they cannot be directly compared. Moreover, the temperature difference is more pronounced for lower $EI_{nvPM}$ ($\sim 2, 5$ K for $EI_{nvPM} = 10^{15}$ kg$^{-1}$, $10^{12}$ kg$^{-1}$) as the number-weighted particle diameter is smaller which increases $S_{v,c}$. This again highlights the importance of considering particle properties when determining the point at which most particles activate (see Sect. 2.3).

## 4.2 Peak plume supersaturation

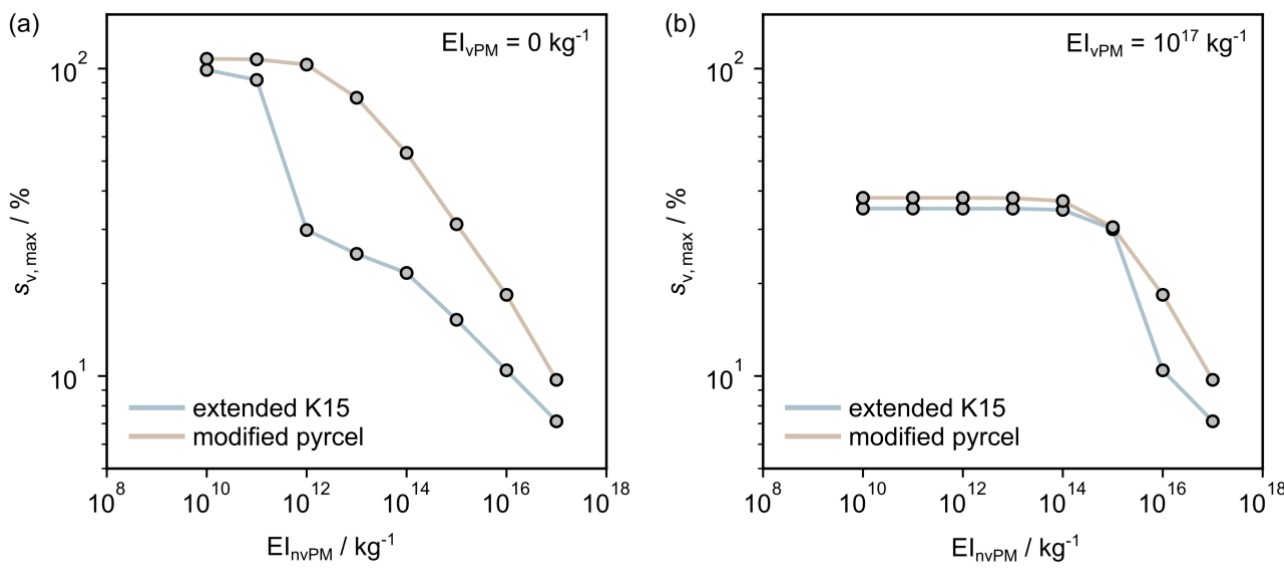

**Figure 5: The maximum plume supersaturation ($s_{v,max}$) across a range of $EI_{nvPM}$ as predicted by the modified pyrcel (brown lines) and extended K15 (blue lines) models. These model predictions have been made (a) without; and (b) with the presence of plume vPM ($EI_{vPM} = 10^{17}$ kg$^{-1}$).**

On average across $EI_{nvPM}$, the maximum plume supersaturations predicted by the extended K15 model are ~10–70% lower than those predicted by the modified pyrcel model, see Fig. 5. Without plume vPM, the largest discrepancies occur when competition between nvPM and ambient particles is largest (at $EI_{nvPM} \sim 10^{12} \, kg^{-1}$, see Fig. 5a). However, when vPM is included, model discrepancies are largest when most contrail ice crystals form via nvPM, see Fig. 5b. Given both pyrcel and the extended

K15 model employ an identical treatment of $P_w$ Eq. (18), these model discrepancies must arise from the treatment of $L_w$. It follows that underestimating the supersaturation at which activation relaxation takes place implies the extended K15 likely overestimates the ability for water droplets to deplete the plume supersaturation. In the final part of Sect. 3.2, we comment on a fundamental limitation of the extended (and original) K15 model that could lead to this discrepancy, which centres around the treatment of $L_w$. In brief, preparing an analytical solution to Eq. (25) demands that (a) the plume supersaturation is

unaffected by particle activation and subsequent droplet growth between adjacent timesteps and (b) the Kelvin effect is discarded from the droplet growth equations. A combination of these effects accounts for the observed behaviour shown in Fig. 5, which underlines the principal limitations of the extended K15 model.

## 4.3 Sensitivity to $EI_{nvPM}$

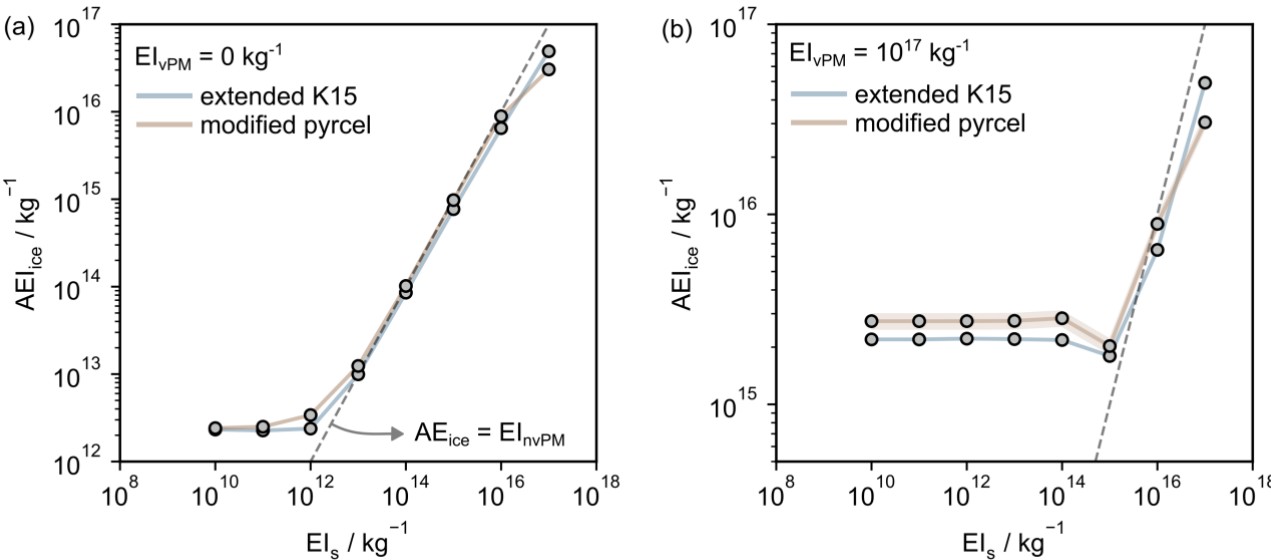

**Figure 6: The dependence of contrail AEI_ice on EI_nvPM as predicted by the modified pyrcel and extended K15 models. Model predictions have been made without (a) and with (b) inclusion of plume vPM.**

In Fig. 6, we show how the ice crystal number concentration scales with $EI_{nvPM}$ in both the extended K15 and modified pyrcel models. This is presented in terms of $AEI_{ice}$, using Eq. (22), so that the results can be more readily compared to other model predictions. Without plume vPM, we find the model agreement for $AEI_{ice}$ is within 35% when $EI_{nvPM} < 10^{16}$ kg$^{-1}$, see Fig. 6a. When plume vPM is included, we find that under this range of $EI_{nvPM}$, differences in model predictions for $AEI_{ice}$ and $s_{v, max}$ exhibit similar trends, see Fig. 5b. Note that $AEI_{ice}$ predictions made using the modified pyrcel model were found to have a maximum uncertainty of ±10%, see SI S11. These derive principally from discretizing the vPM mode, which necessarily approximates the analytic particle size distribution, see SI S4. These errors are shown in Fig. 6 and in future analysis, where appropriate.

For $EI_{nvPM} \geq 10^{16}$ kg$^{-1}$, the extended K15 predicts higher $AEI_{ice}$ than the modified pyrcel model because the extended K15 model does not consider feedback of water vapour between water droplets and ice crystals. To illustrate this effect, suppose we identify a temperature that is warm enough to avoid homogeneous ice nucleation but nonetheless cool enough to promote substantial particle activation, we find that the characteristics of the activated nvPM particles are different in the limit of small and large $EI_{nvPM}$. Note, these limiting values depend on particle size distribution characteristics and properties specified for the contrail mixing line, however in the case of Fig. 6b, they are separated at $EI_{nvPM} \sim 10^{16}$ kg$^{-1}$. To that end, for $EI_{nvPM} \geq 10^{16}$ kg$^{-1}$, the available water vapour in the plume is distributed among many particles and activated nvPM particles typically have values $d_w - d_d < 200$ nm. Comparatively, for $EI_{nvPM} < 10^{16}$ kg$^{-1}$, the nvPM wet particle diameters are distributed quasi-normally and have values $d_w - d_d > 200$ nm. For this reason, the condition for homogeneous ice nucleation is satisfied for each activated nvPM particle for $EI_{nvPM} < 10^{16}$ kg$^{-1}$, since this is sensitive to the relative difference between wet and dry particle diameters, $d_w - d_d$. Hence, we find that the $AEI_{ice}$ is always at least as large as the apparent emission index of activated particles. By comparison, for $EI_{nvPM} \geq 10^{16}$ kg$^{-1}$, given the smaller activated nvPM particles have comparatively low $d_w - d_d$, their size is more sensitive to changes in local supersaturation. Therefore, when the largest activated nvPM particles begin to nucleate ice, this can result in smaller activated nvPM particles evaporating at their expense. In general terms, this is referred to as the Wegener-Bergeron-Findeisen (WBF) process (Korolev, 2007) and provides a rationale for the opposite trend observed in Fig. 6 and Fig. 5 for $EI_{nvPM} \geq 10^{16}$ kg$^{-1}$.

We have previously discussed the importance of vapour feedback between growing droplets and the evolving contrail mixing plume as a source of model agreement, highlighting the coupled nature of growing droplets. These effects are exacerbated by the WBF process, which implies that coupled feedback of water vapour occurring between ice crystals and growing droplets can also contribute to model disagreement. Only the modified pyrcel model is able to describe the kinetic treatment of vapour feedback between hydrometeors and the contrail mixing plume. For this reason, we can place more confidence in model predictions made by the modified pyrcel model than the extended K15 model. Additionally, inclusion of the Kelvin effect in the radial growth equation is only represented in the modified pyrcel model, which is critical for describing water uptake on plume vPM. Hence, we choose to use the modified pyrcel model for the following sensitivity studies.

## 4.4 Sensitivity to ambient temperature

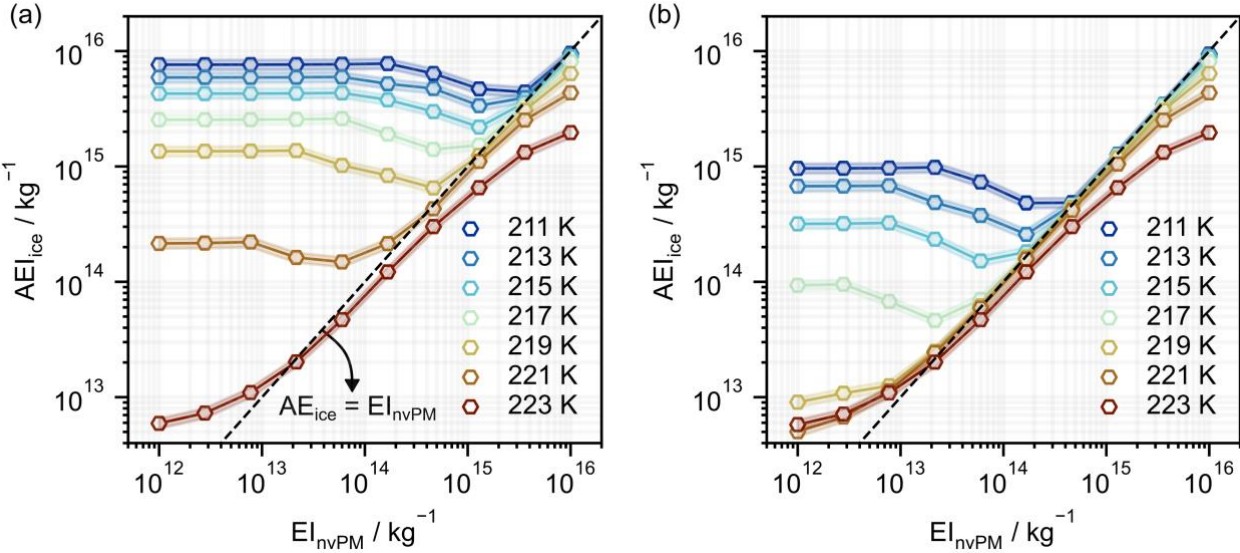

**Figure 7: The dependence of contrail AEI$_{ice}$ on $T_A$ (211 – 223 K, corresponding to $\Delta T_{SAC}$ of 2 – 13 K) and EI$_{nvPM}$ for (a) sulphur-rich and (b) organic-rich plume vPM.**

An evaluation of the modified pyrcel model sensitivity across a range of $T_A$ and EI$_{nvPM}$ revealed several key findings (Fig. 7). Firstly, AEI$_{ice}$ follows a quasi-linear relationship with EI$_{nvPM}$ only when EI$_{nvPM}$ exceeds a temperature-dependent threshold that ranges between $10^{13}$ kg$^{-1}$ and $10^{16}$ kg$^{-1}$. In this quasi-linear region, most ice crystals form via nvPM particles, which defines the "soot-rich" regime. Increasing $T_A$ decreases the peak plume supersaturation so a reduced proportion of the nvPM particles are able to activate (Bräuer et al., 2021), which has a similar effect to decreasing GMD$_{nvPM}$ (see SI S10). The threshold EI$_{nvPM}$ marks the point at which an equal number of contrail ice crystals are formed via nvPM particles and other particles in the plume. Therefore, below this threshold EI$_{nvPM}$, the majority of ice crystals do not form via nvPM particles, which defines the "soot-poor" regime. Owing to the results in Fig. 7, "soot-poor" conditions always satisfy the baseline condition EI$_{nvPM} \leq 10^{12}$ kg$^{-1}$. Here, for warmer $T_A$ ($\sim T_{SAC}$), most contrail ice crystals form via ambient particles and AEI$_{ice}$ approaches an asymptotic limit governed by the assumed ambient particle number concentration. Recall that most ambient particles assumed in our simulations (see, Table. 1) are able to activate at $\Delta T_{SAC} < 1$ K (see, Fig. 3). As $T_A$ is reduced further ($< T_{SAC}$), the increased peak plume supersaturation enables a greater proportion of plume vPM particles to activate (in addition to the ambient particles) and the limiting value of AEI$_{ice}$ increases. Generally, the position of the threshold EI$_{nvPM}$ increases with decreasing $T_A$ as a larger proportion of vPM particles are able to activate and therefore compete with nvPM particles for the plume supersaturation (Yu et al., 2024). These findings suggest that the EI$_{nvPM}$ threshold separating the "soot-rich" and "soot-poor" regime should not be treated as a fixed value, but rather a dynamic threshold that varies with $T_A$.

## 4.5 Sensitivity to plume vPM properties

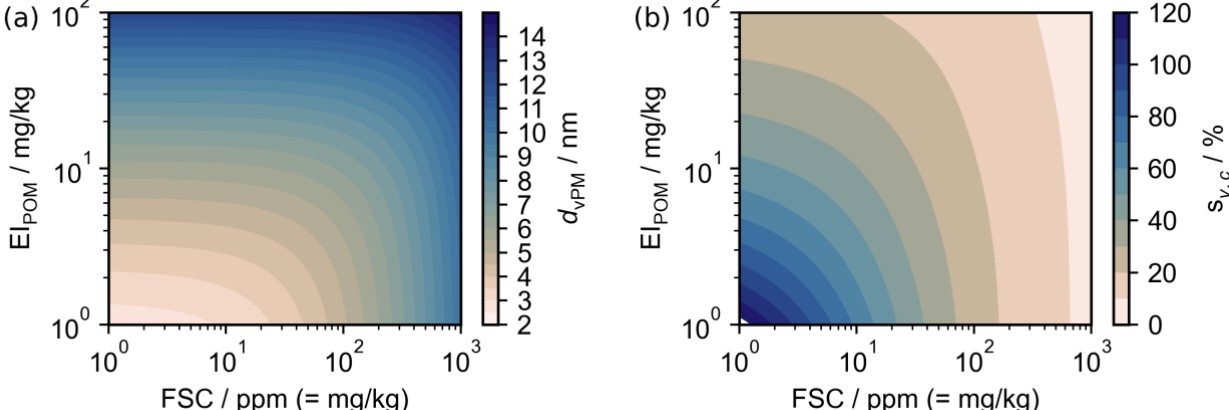

**Figure 8: (a) equilibrium volatile particle modal diameter as a function of fuel sulphur content and the emission index of organic particulate matter, assuming a monomodal distribution and omitting condensation of plume vPM on nvPM, (b) peak plume supersaturation required to activate volatile particles described in (a). We have approximated $EI_{vPM} = 1 \times 10^{17}$, $\varepsilon = 3\%$ and $\rho_{vPM}$ as**
**a mass-weighted sum of the densities of sulphuric acid and POM (1800 kgm$^{-3}$ and 1000 kgm$^{-3}$, respectively), assuming ideal mixing.**

Before continuing with our analysis, we emphasise that $d_{vPM}$ calculated using Eq. (27) provide us with an *upper bound* on the diameter of the volatile mode prior to maximum plume supersaturation; due to the kinetics of particle growth, this diameter is likely reached only after a timescale of minutes-hours (Kärcher et al., 2000). Furthermore, it is possible that some volatile
material may condense on nvPM (if present), which would further reduce $d_{vPM}$. Therefore, the equilibrium plume vPM diameters quoted in Fig. 8a should only be interpreted as upper bounds.

Figure 8a shows that $d_{vPM}$ increases with the $EI_{POM}$ and FSC. For high FSC values (~1000 ppm), $d_{vPM}$ can reach values upwards of 10 nm. By contrast, for low FSC the size becomes limited by POM, without which $d_{vPM}$ reaches 2–3 nm at most. In Fig. 8b, we have used the hygroscopicity parameter for sulphuric acid and POM (assuming $\kappa_{POM} = 0$), to show how
the critical saturation ratio depends on both FSC and POM. For low FSC and $EI_{POM}$, critical supersaturations exceeding 100% are required for particle activation. However, the presence of nvPM typically precludes such high supersaturations (see Fig. 5), therefore given that $GMD_{vPM} \leq d_{vPM}$, volatile contributions to $AEI_{ice}$ are likely low in the "soot-rich" regime. On the other hand, with increasing FSC, plume vPM activation becomes increasingly likely as there are fewer nvPM particles available to quench the plume supersaturation.

As discussed, the results presented above are valid only for providing insight into the approximate equilibrium behaviour of plume vPM formation. Within a real contrail mixing plume, the time available for volatile particle growth by accretion of POM and/or sulphuric acid is reduced and kinetics become important. To that end, the dependence of volatile

mode characteristics on POM (including lubrication oil) and FSC have previously been investigated in simulations performed at cruise (Cantin et al., 2025; Rojo et al., 2015) and ground level (Jones and Miake-Lye, 2024; Wong et al., 2014, 2015). Next, we investigate the dependence of contrail $AEI_{ice}$ on prescribed vPM properties in the modified pyrcel model

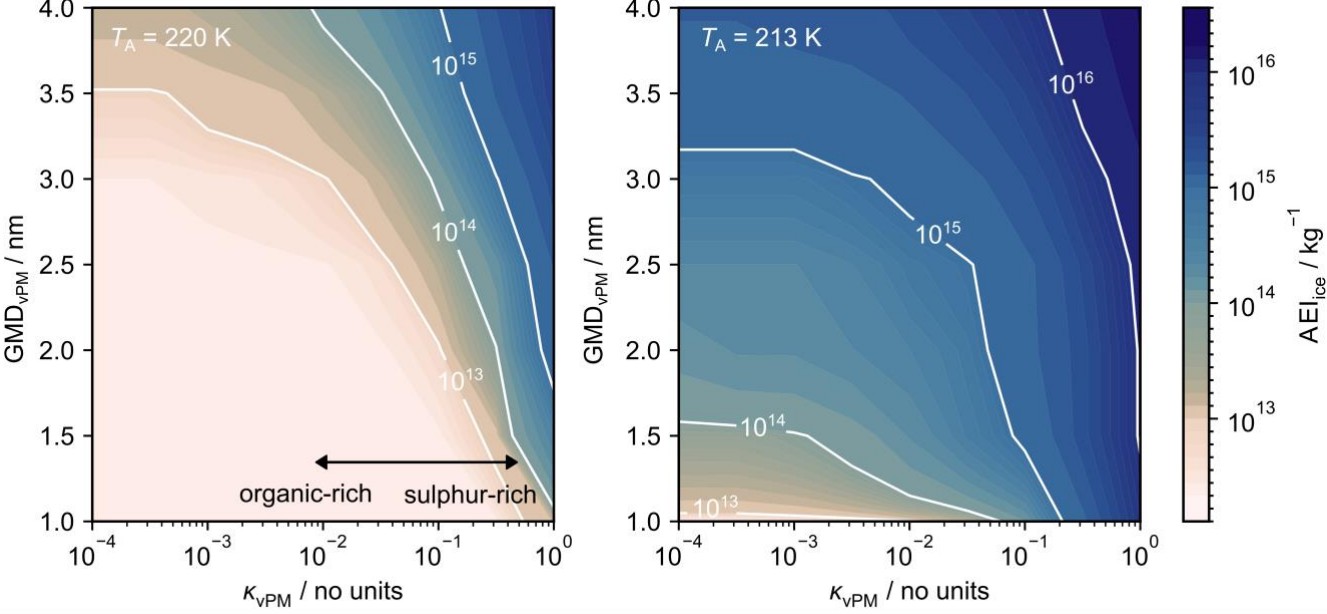

**Figure 9: The dependence of contrail $AEI_{ice}$ on the plume vPM properties ($GMD_{vPM}$ and $\kappa_{vPM}$) at ambient temperatures of: (a) 220 K; and (b) 213 K. For both scenarios, the assumed particle properties are outlined in Table 1, $EI_{vPM}$ and $EI_{nvPM}$ are assumed to be $10^{17}$ kg$^{-1}$ and $10^{12}$ kg$^{-1}$ respectively (in line with the "soot-poor" baseline conditions described in Sect. 4.4) and the contrail $AEI_{ice}$ is simulated using the modified pyrcel model with model time steps of 0.1 ms and a maximum duration of 0.5 s.**

Figure 9 demonstrates the sensitivity of the modified pyrcel model to the properties of the plume vPM mode in the "soot-poor" regime, subject to our assumptions on the hygroscopicity of the condensable POM. Our results suggest that a sulphur-rich plume vPM mode can lead to $AEI_{ice} > 10^{15}$ kg$^{-1}$, even when $GMD_{vPM} < 2$ nm. Conversely, when the plume vPM mode is organic-rich, the $AEI_{ice}$ only rises above $10^{15}$ kg$^{-1}$ under extremal conditions (i.e., low $T_A$ and high $GMD_{vPM}$). For most other cases, $AEI_{ice} \ll 10^{15}$ kg$^{-1}$ and the organic-rich plume vPM mode competes with ambient particles for plume supersaturation. Therefore, if soot particle emissions could be reduced in the vicinity of $10^{12}$ kg$^{-1}$, provided the volatile particle mode is largely composed of sulphuric acid, we anticipate only a small reduction in $AEI_{ice}$ values relative to "soot-rich" exhausts. On the other hand, if the volatile particle mode is largely composed of organic material, we find that $AEI_{ice}$ values could be reduced by a factor of up to 100, with benefits ultimately constrained by the size of the plume vPM particles and the properties of entrained ambient particles.

### 4.6 Sensitivity to aircraft parameters

The final sensitivity analysis will focus on the gradient of the SAC contrail mixing line (Kärcher et al., 2015), which contains important information about aircraft and fuel properties, see Sect. 2.1. Simultaneously, we will explore the sensitivity to ambient temperature, so that the resulting output, $AEI_{ice}(G, T_A)$, provides a useful reference for estimating $AEI_{ice}$ for various combinations of aircraft properties and ambient conditions under "soot-poor" conditions, as defined in Sect. 4.4. For both these scenarios, a sulphur-rich volatile mode was prescribed, as in Sect. 3.4.

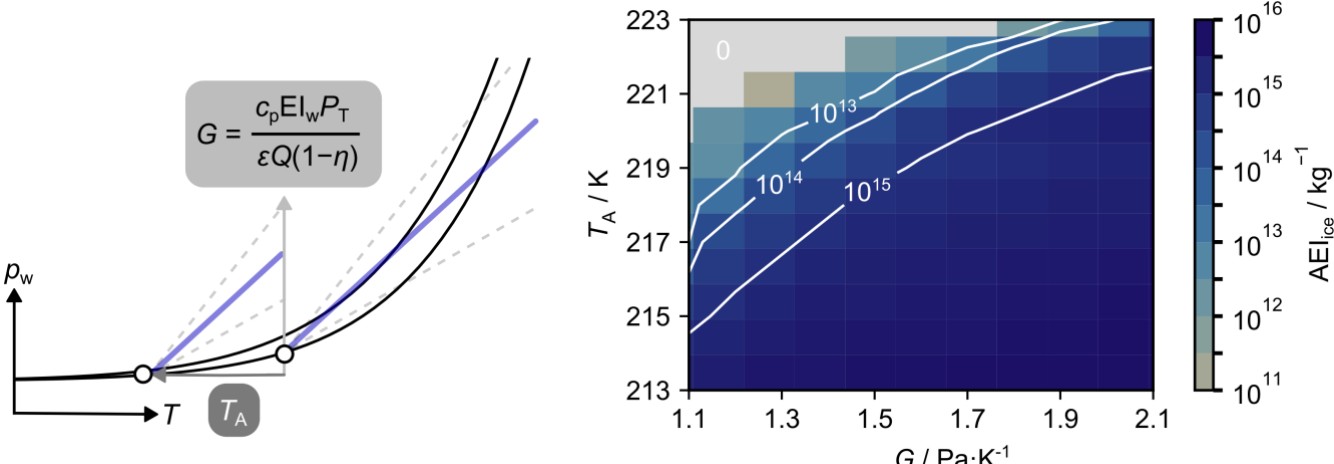

**Figure 10: The dependence of contrail AEI$_{ice}$ on $T_A$ and the gradient of the contrail mixing line, as defined in Eq. (1). The simulation has been performed for sulphur-rich plume vPM and EI$_{nvPM}$ of $10^{12}$ kg$^{-1}$, in line with the "soot-poor" baseline conditions described in Sect. 4.4. The region shaded in red corresponds to AEI$_{ice}$ = 0. Physically this represents mixing lines that do not intersect the water saturation line.**

Figure 10 shows how the $AEI_{ice}$ varies as a function of $G$ and $T_A$ for "soot-poor" baseline conditions as described in Sect. 4.4 and sulphur-rich plume vPM. For a fixed ambient temperature, increasing $G$ leads to an increased peak plume supersaturation and hence an increase in $AEI_{ice}$. For ambient temperatures with $\Delta T_{SAC} > 10$ K (here, $T_A > 215$ K), enhancements of up to three orders of magnitude are observed within the prospective range of $G$ values for kerosene-based fuel (1.1 – 2.2). This aligns with

results from the original SAC framework, namely that parameters such as overall efficiency (Schumann, 2000), cruise altitude (Gryspeerdt et al., 2024) and water vapour emissions are critical for estimating resulting contrail ice crystal properties, and hence the radiative impact of contrail-cirrus.

## 5. Implications

Here, we contextualise several of the key findings from Sect. 4 for global contrail simulations (Sect. 5.1), contrail mitigation (Sect. 5.2) and engine and fuel technologies (Sect 5.3).

### 5.1 Global contrail simulations

Several contrail simulations rely on nvPM (and ambient particle) properties to parameterize $AEI_{ice}$ in young contrails (Bier and Burkhardt, 2022; Teoh et al., 2024). In the "soot-poor" regime, we find that including activation of plume vPM might enhance $AEI_{ice}$ by up to three orders of magnitude. Increased $AEI_{ice}$ is correlated with contrail RF (Burkhardt et al., 2018), implying that these current global contrail simulations may underestimate global contrail forcing. While the extended K15 and modified pyrcel models are both too computationally demanding to be directly integrated into global contrail simulations, we suggest that either model could be incorporated in these simulations by preparing multidimensional lookup tables. Importantly, as the $AEI_{ice}$ predictions are sensitive to the assumed properties of plume vPM and these are highly uncertain at present, a global contrail simulation relying on the extended K15 model would serve only as a sensitivity study.

### 5.2 Contrail mitigation

Operational contrail mitigation has hitherto focussed on circumventing regions where $s_i > 0$ in order to mitigate contrail *persistence*. Our simulations imply that under "soot-poor" conditions and for flights within 5 K of $T_{SAC}$, increasing the ambient temperature by 2 K may reduce $AEI_{ice}$ by up to a factor of 10 (see Fig. 9). If this behaviour could be experimentally validated, an analogous strategy might be possible by circumventing cooler regions of the upper troposphere, to mitigate contrail *formation* under "soot-poor" conditions. Moreover, maximum predictive biases between the European Centre for Medium-Range Weather Forecasts (ECMWF) ERA5 atmospheric reanalysis model and In-service Aircraft for A Global Observing System (IAGOS) observations of $T_A$ and $s_i$ are -0.2% and -5.5%, respectively (Wolf et al., 2025), which implies that atmospheric requirements for contrail *formation* can be predicted more accurately than for contrail *persistence*. Assuming an upper tropospheric lapse rate of -6.5 K/km per the ICAO standard atmosphere (ICAO, 1993), a 2 K temperature difference corresponds to approximately 10 flight levels (1000 ft/305 m). This is equivalent to the reduced vertical separation minima between aircraft in the FL 290 - FL 410 flight band (ICAO, 2012), which implies that it might be possible to adjust $T_A$ by making relatively small changes in elevation. Using $T_A$ data from ECMWF, we have explored this further in SI S12. Nevertheless, avoidance of local cold-spots would likely involve reducing aircraft cruise altitudes, consequently increasing (a) the likelihood for aircraft emissions to perturb clouds in the mixed-phase regime, (b) the number of turbulence events (Dehghan et al., 2014) and (c) direct emissions of $CO_2$. Therefore, we reiterate that this approach is subject to both inherent model uncertainties and the aforementioned concerns (a) – (c), which need to be resolved before exploring this further.

**5.3 Engine and fuel technologies**

Aircraft powered by lean-burn combustors have $EI_{nvPM}$ that range between $10^{10}$ and $10^{12}$ kg$^{-1}$ (ICAO, 2024), which coincides with our definition for the "soot-poor" regime given in Sect. 4.4. Under these conditions, $AEI_{ice}$ is controlled by the properties of plume vPM emissions and $T_A$ (see Fig. 9). Therefore, $AEI_{ice}$ can be reduced practically by (a) avoiding cooler ambient temperatures (see Sect. 5.1) and (b) reducing the mass of condensable volatile material. To that end, we suggest the removal of lubrication oil emissions and diminution of FSC. In Sect. 4.4, we demonstrate that FSC scales with the mass of condensable sulphuric acid. Therefore, for fixed $EI_{vPM}$, increasing FSC may produce plume vPM particles with higher $GMD_{vPM}$ and higher $\kappa_{vPM}$. This intuition agrees with results collated across various flight campaigns, showing that $EI_{vPM}$ for particles larger than 5 nm increase with increasing FSC (Schumann et al., 2002). In the "soot-poor" regime, $AEI_{ice}$ and hence the potential contrail climate impact increase with $GMD_{vPM}$ and $\kappa_{vPM}$. Therefore, we would advise that FSC is reduced as low as practicably possible, as FSC levels of several ppm (assuming $EI_{POM} = 0$ kg$^{-1}$) are sufficient to produce plume vPM particles with $d_{vPM}$ on the order of several nm (see Sect. 4.5). We note that comparable FSC reductions have been successfully demonstrated in other sectors (International Maritime Organization, 2020) and that the largest benefits would be achieved if low FSC fuels were used in lean-burn engines, where $AEI_{ice}$ values are most sensitive to the properties of plume vPM emissions.

Finally, measurements show that SAF can reduce both $EI_{nvPM}$ and $GMD_{nvPM}$ relative to conventional fuel (Moore et al., 2017; Voigt et al., 2021). Under conditions where most contrail ice crystals form via nvPM particles, both changes would reduce $AEI_{ice}$ as fewer particles are able to activate at a given supersaturation. These effects have previously been reported using contrail parcel model simulations to compare with ECLIF (Emission and CLimate Impact of alternative Fuels) measurements (Yu et al., 2024). Additionally, for variable $EI_{nvPM}$ and fixed plume vPM properties, decreasing $GMD_{nvPM}$ would increase the position of the minimum in Fig. 9 (see SI S10).

**6. Conclusions**

Here, we extended two parcel models to describe particle microphysics during the early stages ($t < 1$ s) of contrail plume evolution. Firstly, the minimal framework originally developed by (Kärcher et al., 2015) was extended to account for plume vPM by incorporating an additional particle mode. We replaced several analytical approximations with numerical solutions and increased the accuracy of others, so that the model could be more applicable for describing the activation of plume vPM. Secondly, using similar parameterizations for the mixing process but a higher-fidelity description of $L_w$, a cloud parcel model (pyrcel) (Rothenberg and Wang, 2016) was modified to treat contrail formation and used as a benchmark for model comparison.

We found the extended K15 model systematically underpredicted the peak plume supersaturations when compared to the modified pyrcel model. This is likely because it does not incorporate the Kelvin effect and therefore overpredicts droplet growth rates. Generally, the estimated $AEI_{ice}$ from both models are within 35% for $EI_{nvPM} \leq 10^{16}$ kg$^{-1}$, when activation of plume vPM is excluded. However, when plume vPM activation is included, the modified pyrcel model predicts higher ice

crystal number concentrations by 10-30% for $\text{EI}_{\text{nvPM}} \leq 10^{13}$ kg$^{-1}$. Discrepancies are more pronounced for $\text{EI}_{\text{nvPM}} > 10^{13}$ kg$^{-1}$ due to differing treatment of kinetics and water vapour competition between hydrometeors between models.

Using the modified pyrcel model, we performed several sensitivity studies and found that across $\text{EI}_{\text{nvPM}}$, the $\text{AEI}_{\text{ice}}$ decreases with decreasing peak plume supersaturations as predicted by the original SAC mixing line. This is controlled by local ice supersaturation ($s_i$), ambient relative humidity, $T_A$ and the gradient of the mixing line. Therefore, decreasing the overall efficiency and water vapour emissions results in fewer contrail ice crystals being produced. Also, we find that the threshold between "soot-rich" and "soot-poor" conditions (as defined in Sect. 4.4) is sensitive to the properties of plume vPM ($\text{EI}_{\text{vPM}}$, $\text{GMD}_{\text{vPM}}$, $\kappa_{\text{vPM}}$) and $T_A$, and therefore should be treated as a dynamic threshold. Notably, we find that the threshold $\text{EI}_{\text{nvPM}}$ increases with decreasing $T_A$ as an increasing proportion of vPM particles are able to compete with nvPM particles for the plume supersaturation. In the "soot-poor" regime, including plume vPM activation leads to $\text{AEI}_{\text{ice}}$ enhancements by up to three orders of magnitude; these are maximised if the material is highly hygroscopic (e.g., sulphur-rich).

As the aviation industry reduces $\text{EI}_{\text{nvPM}}$ and therefore approaches the "soot-poor" regime, we find that existing contrail-cirrus models that do not extend to include plume vPM activation will underestimate $\text{AEI}_{\text{ice}}$ and therefore global climate forcing. Future research should be directed towards (i) ground-based and *in-situ* measurements focussing on the "soot-poor" regime and its transition towards the "soot-rich" regime, with near-field characterisation of *both* nvPM and plume vPM properties, (ii) laboratory measurements of the activation properties and/or ice nucleating ability of particles representative of plume vPM present in nascent contrail exhausts and (iii) validation of plume aerosol and contrail formation models (parcel model/LES simulations) using *in-situ* observations.

**Notation**

| | |
|---|---|
| SAC | Schmidt-Appleman Criterion |
| nvPM | Non-volatile particulate matter |
| vPM | Volatile particulate matter |
| $\text{EI}_{\text{vPM}}$, $\text{EI}_{\text{nvPM}}$ | Number emission index of vPM, nvPM |
| SAF | Sustainable aviation fuel |
| GMD | Geometric mean diameter |
| GSD | Geometric standard deviation |
| K15 | Microphysical model developed by Kärcher et al (Kärcher et al., 2015) |
| $T_E$ | Gas temperature at the engine exit plane |
| $T_A$ | Ambient temperature |
| $T$ | Plume temperature |

| | |
|---|---|
| $G$ | Gradient of average contrail mixing line |
| $p_{v,M}$ | Partial pressure of water vapour in the contrail mixing plume |
| $p_{v,E}$ | Partial pressure of water vapour at the engine exit plane |
| $p_{v,A}$ | Partial pressure of water vapour in the ambient environment |
| $c_p$ | Isobaric specific heat capacity of air |
| $EI_w$ | Mass emission index of water vapour |
| $P_T$ | Total air pressure |
| $\varepsilon$ | Ratio of gas constants for water vapour and dry air |
| $Q$ | Total heat released per mass of fuel burned |
| $\eta$ | Overall aircraft efficiency |
| $p_{liq}, p_{ice}$ | Saturation vapour pressure above a plane surface of supercooled water, ice |
| $S_{v,M}, S_{i,M}$ | Water, ice saturation ratio in a contrail mixing plume |
| $s_{v,M}$ | Water supersaturation in a contrail mixing plume |
| $T_{SAC}$ | SAC temperature |
| $\Delta T_{SAC}$ | Difference between the ambient and SAC temperatures ($T_{SAC}$ - $T_A$) |
| $S_v$ | Water saturation ratio above a liquid droplet |
| $a_w$ | Water activity |
| $\sigma_s$ | Surface tension at the droplet/air interface |
| $M_w$ | Molar mass of water |
| $\rho_w$ | Density of water |
| $R$ | Global gas constant |
| $d_w$ | Wet particle diameter |
| $d_d$ | Dry particle diameter |
| $\kappa$ | Hygroscopicity parameter |
| $d_{w,c}, r_{w,c}$ | Critical wet particle diameter, radius |
| $S_{v,c}$ | Critical water saturation ratio |
| $w_v$ | Water vapour mixing ratio |
| $w_c$ | Mixing ratio of condensed water vapour |
| $P_w$ | Water (super)saturation production rate in the absence of particles |
| $L_w$ | Water (super)saturation depletion rate due to particle activation and droplet growth |
| $\beta$ | Plume dilution parameter |

| | |
|---|---|
| $D$ | Plume dilution factor |
| $\tau_m$ | Timescale over which contrail mixing is unaffected by entrainment |
| $\mathcal{N}_0$ | Mass-based mixing factor |
| $n_T$ | Total number concentration of aircraft-mode particles |
| $n_{a,T}$ | Total number concentration of ambient-mode particles |
| $n_{a,0}$ | Ambient particle number concentration |
| $\rho_a$ | Mass density of ambient air |
| $\dot{r}_w$ | Radial droplet growth rate |
| $n_w$ | Droplet number concentration |
| $\dot{n}_w$ | Rate of change of droplet number concentration |
| FSC | Fuel sulphur content |
| POM | Particulate organic matter |
| $EI_{H_2SO_4}$ | Mass emission index of sulphuric acid |
| $EI_{POM}$ | Mass emission index of POM |
| $\varepsilon_s$ | Conversion efficiency of sulphur to sulphuric acid |
| $d_{vPM}$ | vPM equilibrium diameter |
| $\rho_{vPM}$ | Density of condensed vPM |

**Competing Interests**

The authors declare that they have no conflict of interest.

**Code and data availability**

Code for the modified pyrcel model and extended K15 model is available at the following addresses: https://zenodo.org/records/16901327 and https://zenodo.org/records/16901290.

**Author contribution**

JP and MEJS conceptualized the project. JP developed the software, conducted the initial investigation and wrote the original manuscript. All the authors developed the methodology and edited the manuscript. MEJS supervised the project and acquired funding.

## Acknowledgements

We would like to thank Daniel Rothenberg for helpful discussions on revising pyrcel to describe contrail formation, Fangqun Yu for his insight on the model description of plume vPM, and Christiane Voigt for useful discussions on model comparison with *in-situ* measurement data. Finally, we would like to thank Yu Wang for providing the ECHAM-HAM data used in SI S5.

## Financial support

This research has been supported by the Engineering and Physical Sciences Research Council (grant no. EP/S023593/1).

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
