# Peer review of "An updated microphysical model for particle activation in contrails: the role of volatile plume particles"

_EGUsphere, 2025_

## Referee Comment (RC2)

**An updated microphysical model for particle activation in contrails : the role of volatile particles**

**Overall**: This paper update two different microphysical model and perform sensitivity analysis to input parameters with only one of this model, the one that seems the most realistic. Their update consist in adding a distribution to account for volatile particles in order to be able to compute contrail formation in low soot regime. Overall the model modification are well explained as well as most of the underlying hypothesis; however several informations are missing such as what is the density of the volatiles particles, the surface tension… The number of volatile particle is fixed and their appearance process is not model. This limitation is clearly mentioned. But I regret the lake of sensitivity analysis on this parameter of the model.

Major

The paragraph 4.4 is interesting. This kind of graph have already been made by https://doi.org/10.1021/acs.est.4c04340 and https://doi.org/10.21203/rs.3.rs-6559440/v1 how do your graph compare to these papers?

Line 592-595, you status that volatile mode characteristics on POM and FSC has not been talked at curise altitude. However Rojo et al 2015 clearly include POM in their volatile and they perform a sensitivity analysis to FSC and POM emission index on contrail formation.

In line 577-581 you underline the impossibility of dvPM being found by equation 22 but you tell in line 459 that you use it to estimate this diameter. Therefor I m a bit lost. Do you use this formula or not? If yes, what is the error made by this choice?

I will assume in this comment that you use the formula 22 to choose the initial diameter of the volatile particles. In order to find it, you need to define the density of the volatile particles. However, you give no indication of its value. Please provide this information. The same remark can be made for the surface tension, what value do you use?

You perform a sensitivity analysis on the volatile particle emission index by doing 0 or 1e17. Since this is a major hypothesis of your work, I suggest a more detailed sensitivity analysis such as EI=1.e16-18 for example.

In the acknowledgement you mention Christiane Voigt for "model comparison with in situ measurement". If you have access to experimental data, why don't you show some experimental validation of your model?

For what I understand of the model, every volatile particles are a mixture of organics and sulfuric acid. Considering that the organics are mainly insoluble species and sulfuric acid is soluble, is there no possibilities to have in fact two different kind of particles one of pure sulfuric acid and one of pure organics?

Minor:

Line 45-48: You give an experimental definition of volatile particles which is right, however the reader may be confuse if these particles exist or not at the exit of the engine. I suggest adding a sentence, which says that considering the exit temperature of the engine, the volatile particles forms during the cooling of the plume.

Line 50: you give a value for the apparent emission index of volatile particles of 1E17 kg-1. In a recent paper (still a pre-print) https://doi.org/10.21203/rs.3.rs-6559440/v1 they show inflight measurement. The total particle number is limited between 5E14 and 5E15, depending of the flight condition and fuel used. Since this is an order of magnitude lower, it would be great to add more references from the literature.

Line 75-80: I suggest you to read https://doi.org/10.1080/02786826.2024.2395940 and https://doi.org/10.1016/j.jaerosci.2025.106612 which introduce in complex CFD a microphysical model which seems close to the one from Wong 2014 model.

Line 226-227: "However, as outline in SIS3, this criterion cannot be reconciled with minimum requirement for particle activation" is a bit too negative since particles may be already big enough to be activated has shown by line 245-251. I suggest to restrict this statement for small particles.

Line 270: you neglect the uptake of water of the atmosphere while you take into account the uptake of ambient particle. I agree that at first it is negligible but at the end it will probably influence the size of the ice crystals. Considering the low cost of such modification I don't really understand why have you done this choice.

In equation 16 you neglect the variation of temperature due to freezing and condensation whereas in the Pyrcel website it is included. Can't it also be included in K15? Moreover in the Pyrcel description in lines 407-412 you

replace the gravity term by the one given in this equation, do you keep the latent heats term?

In line 450-453 you distinguish between oil organics and gaseous emission. In line 461 I have the impression that you treat them as the same species. Are they treated the same way?

You conclude 4.3 by telling that Pyrcel is better than K15 but in line 554 you tell that you make the sensitivity analysis with K15. I guess you have use Pyrcel but it is just a typo.

In line 645 you say that the K15 model could be incorporated in global contrail simulations, however considering the comparison with Pyrcel, I wonder why K15 and not Pyrcel?

In Supplementary material S1, figure S1: please recall the k value used in order to simplify the reading.

Limitation given in line 351-354 has to be emphasis in the introduction since it is an important one.

In line 54 of the supplementary material, you give the probability to transform into ice. Then in line 56 you say that the particle freeze when the probability is one. However, the line 54 formula shows that this probability is never equal to one. Then you have to choose a threshold. You have to give the information to the reader.

---

## Author Comment (AC1)

**Response to Reviewer Comments**

We thank the reviewers for their comments, which have helped us to improve the quality of the manuscript. In this document, reviewer comments are indicated using italic text. Our responses are written using normal (non-italic) text. When page and line numbers are given (in red), these refer to the original manuscript unless otherwise stated. Standard blue text is used to indicate text cited from the revised manuscript and **bold blue text** is used to indicate new material introduced into the manuscript.

**Referee 1 (RC1)**

**Main comment**

1.  *Evaluation against observations: It would be good to assess the extent to which the inclusion of vPM activation (using the framework proposed here) affects simulated AEI_ice for some case studies where observational data are available, e.g. PAZI-2, ECLIF II, CONCERT – see Table 1 in (Bier and Burkhardt, 2022).*

    - Thank you for your suggestion. We agree that it would be beneficial to compare our model outputs to available *in-situ* data from measurements campaigns.
    - While we largely focussed on exploring underlying model sensitivities in this work, we will address *in*-situ comparisons in an upcoming manuscript using our extended K15 model in combination with Boeing ecoDemonstrator and NEOFUELS/VOLCAN measurements (Teoh et al., [in preparation]).

**Minor comments**

2.  *Section 5.1 needs to articulate more clearly the extent to which current global contrail simulations are impacted by the limitations addressed by this proposed new scheme. The current text implies that all existing global contrail models rely on nvPM properties to parameterise AEI_ice, leading to the conclusion that "current global contrail simulations may underestimate global contrail forcing" (line 643). This is not necessarily true, as there are significant differences in how existing contrail schemes in climate models initialize contrails.*

Thank you for your suggestion. You are correct to highlight that there are significant differences between contrail initialization schemes. For example:

- In ECHAM-HAM-CCMod (Bier and Burkhardt, 2022), contrails are initialized according to (Kärcher et al., 2015) (the K15 model without our proposed extensions). Here, $AEI_{ice}$ depends on ambient conditions, $EI_{nvPM}$ and the concentration of ambient particles.
- In CoCiP (Teoh et al., 2024), initial contrail $AEI_{ice}$ is estimated by scaling $EI_{nvPM}$ using a parameter that depends on the temperature difference $T_A$ - $T_{SAC}$. Both these models rely on $EI_{nvPM}$ to parameterize $AEI_{ice}$, which implies that incorporation of vPM would increase $AEI_{ice}$ and therefore global contrail forcing.
- However, as you have underlined, there are models that do not initialize $AEI_{ice}$ according to $EI_{nvPM}$. For example, in CAM5 (Chen and Gettelman, 2013), the contrail "parameterization does not consider direct participation of aviation [nvPM] during the formation process of contrails which could highly affect the ice particle size and number concentration of fresh contrails" (Chen and Gettelman, 2016). Therefore, in this case, their model would be unaffected by changes to the properties of particles entrained in the mixing plume.

Considering these model differences, we have clarified that our conclusions are applicable to only a subset of global climate models:

[main text: lines 675 – 678]:

"**Several g**Global contrail simulations **rely** on nvPM properties to parameterize $AEI_{ice}$ in young contrails (Bier and Burkhardt, 2022; Teoh et al., 2024). In the "soot-poor" regime, we find that including activation of plume vPM might enhance $AEI_{ice}$ by up to three orders of magnitude. Increased $AEI_{ice}$ is correlated with contrail RF (Burkhardt et al., 2018), implying that **these** global contrail simulations may underestimate global contrail forcing."

3. *Lines 9-11: While the study suggests that including vPM activation in models is likely to increase contrail cirrus RF estimates, the use of "however" in the second sentence of the abstract could be interpreted to imply the opposite. I recommend rephrasing this to avoid potential confusions.*

Thank you for your suggestion. We agree that the use of "however" may be misinterpreted so have modified this sentence as such:

[main text: lines 10 – 11]:

"**Currently**, these simulations assume non-volatile particulate matter (nvPM) and ambient particles are the only source of condensation nuclei, omitting activation of volatile particulate matter (vPM) formed in the nascent plume."

4. *The phrase "both models" on line 16 of the abstract is unclear, as the two models are not introduced beforehand. Consider revising the opening sentences to explicitly mention the models being compared, which will help orient the reader.*

Thank you for your suggestion. We agree that this was unclear so have modified lines 11 – 12 in the abstract to explicitly define the two models:

[main text: lines 15 – 16]:

"Here, we extend a microphysical **model** to include vPM and benchmark this against a **more advanced** parcel model (pyrcel) modified to treat contrail formation."

**Technical Corrections**

5. *Line 230: at least on my screen what is referred to as green lines appear in fact as grey.*

Thank you for your suggestion. We have viewed this document on different devices and agree that there is some ambiguity regarding the line colour. To resolve this, we choose to refer to this colour as "grey/green", which we find is sufficiently distinct from the purple line in Figure 3a. We have therefore made the following changes in the manuscript:

[main text: Fig. 3 caption]

"Figure 3: (a) Saturation vapour pressure above water ($p_{\text{liq}}$) (purple, dotted) and a contrail mixing line (purple, solid) for threshold conditions, $\Delta T_{\text{SAC}}$ = 0 K. Particle properties ($d_{\text{d}}$, $\kappa$) have been chosen so that the modified saturation vapour pressure above water $p_{\text{liq}} \cdot S_{\text{v,c}}$ ($d_{\text{d}}$, $\kappa$) (**grey**/green, dotted) translates threshold conditions by 10 K, so $\Delta T_{\text{SAC}}$ = 10 K (**grey**/green, solid). (b) Extension of panel (a) to show dependence of $\Delta T_{\text{SAC}}$ on $d_{\text{d}}$ and $\kappa$. Combinations of particle properties resulting in $\Delta T_{\text{SAC}}$ = 10 K have been shown (**grey**/green, solid). In all cases, $S_{\text{v,c}}$ is estimated conservatively using $T$ = 220 K."

[main text: lines 238 - 240]

"Figure 3a shows that when these particle properties are included, the threshold requirement for activation (previously water saturated conditions, black dotted line) is elevated (**grey**/green dotted line)."

6. *Line 550: Fig. 7 caption should mention 211-223K rather than 211-213K.*

Thank you for noticing this. We have made this correction in the original text.

**Referee 2 (RC2)**

**Overall comment**

- *This paper update two different microphysical model and perform sensitivity analysis to input parameters with only one of this model, the one that seems the most realistic. Their update consist in adding a distribution to account for volatile particles in order to be able to compute contrail formation in low soot regime. Overall the model modification are well explained as well as most of the underlying hypothesis; however several informations are missing such as what is the density of the volatiles particles, the surface tension… The number of volatile particle is fixed and their appearance process is not model. This limitation is clearly mentioned. But I regret the lake of sensitivity analysis on this parameter of the model.*

**Major comments**

1. *The paragraph 4.4 is interesting. This kind of graph have already been made by https://doi.org/10.1021/acs.est.4c04340 and https://doi.org/10.21203/rs.3.rs-6559440/v1 how do your graph compare to these papers?*

- $AEI_{\text{ice}}$ estimates from the extended K15 model are generally consistent with the trends predicted by these studies (quoted above). For example, linear trendlines for high $EI_{\text{nvPM}}$, and negative correlations with temperature and $EI_{\text{nvPM}}$ (under soot-poor conditions.
- We have compared the $AEI_{\text{ice}}$ estimates from the extended K15 model with measurements from the Boeing ecoD and NEOFUELS/VOLCAN campaigns. These will be described in a separate upcoming study (Teoh et al., [in preparation]), as mentioned in our response to RC1 (1) above.

2. *Line 592-595, you status that volatile mode characteristics on POM and FSC has not been talked at curise altitude. However Rojo et al 2015 clearly include POM in their volatile and they perform a sensitivity analysis to FSC and POM emission index on contrail formation.*

Thank you for bringing this to our attention. We had mistakenly overlooked these sentences after introducing the (Rojo et al., 2015) reference into the manuscript introduction. We have corrected this in the original text as follows: (note, due to the large number of inline changes, we have shown a before/after below)

[main text: lines 592 - 595]

Before:

"To our knowledge, the dependence of volatile mode characteristics on POM (including lubrication oil) and FSC has not been investigated for cruise altitude, although the significance has been elucidated in models of ground-based volatile particle growth (Jones and Miake-Lye, 2024; Wong et al., 2014, 2015)."

After:

**"To that end, the dependence of volatile mode characteristics on POM (including lubrication oil) and FSC have previously been investigated in simulations performed at cruise (Cantin et al., 2025; Rojo et al., 2015) and ground level (Jones and Miake-Lye, 2024; Wong et al., 2014, 2015). Next, we investigate the dependence of contrail AEI$_{ice}$ on prescribed vPM properties in the modified pyrcel model."**

3. *In line 577-581 you underline the impossibility of dvPM being found by equation 22 but you tell in line 459 that you use it to estimate this diameter. Therefor I m a bit lost. Do you use this formula or not? If yes, what is the error made by this choice?*

4. *I will assume in this comment that you use the formula 22 to choose the initial diameter of the volatile particles. In order to find it, you need to define the density of the volatile particles. However, you give no indication of its value. Please provide this information. The same remark can be made for the surface tension, what value do you use?*

- Regarding (3): in the final sensitivity analysis (Fig. 7, Fig. 9 and Fig. 10), we prescribe the plume vPM properties according to the values given in Table. 1. These are estimated at the timescale of "particle activation and droplet growth in the nascent contrail mixing plume (~ 1 s)" (lines 443 – 444) using references given inline and in the caption of Table. 1. Separately, we also estimate projected equilibrium properties for plume vPM, that would be achieved after much larger timescales, and if contrail formation did not take place. We make the distinction between plume vPM properties relevant for contrail formation and equilibrium properties in (line 444). Regarding (4), Eq. (22) is therefore used to estimate the size of equilibrium vPM. Equation (22) is not used to prescribe the size of plume vPM relevant for contrail formation as this property is taken from Table. 1. To clarify this distinction, we have introduced the following change:

[main text: lines 443 - 445]

"These are the characteristics at the timescale of particle activation and droplet growth in the nascent contrail mixing plume (~ 1 s); however,. **Separately,** we can also estimate the (maximum) equilibrium plume vPM diameters ($d_{vPM}$) that would be reached at t >> 1 using a mass balance."

- We have also noticed that we mistakenly referred to the equilibrium vPM diameter ($d_{vPM}$) rather than the prescribed vPM diameter ($GMD_{vPM}$) in several locations in Sect 4.5, which we have now resolved.

[main text: lines 597 – 598 and Fig. 9 axes labels]

"Figure 9: The dependence of contrail AEIice on the plume vPM properties ($d_{vPM}$$GMD_{vPM}$ and $\kappa_{vPM}$) at ambient temperatures of: (a) 220 K; and (b) 213 K."

[main text: lines 602 – 605]

"Figures 8 and 9 demonstrate the sensitivity of the modified pyrcel model to the properties of the plume vPM mode in the "soot-poor" regime, subject to our assumptions on the hygroscopicity of the condensable POM. Our results suggest that a sulphur-rich plume vPM mode can lead to AEI$_{ice}$ > $10^{15}$ kg$^{-1}$, even when **GMD**$d_{vPM}$ < 2 nm. Conversely, when the plume vPM mode is organic-rich, the AEI$_{ice}$ only rises above $10^{15}$ kg$^{-1}$ under extremal conditions (i.e., low $T_A$ and high **GMD**$d_{vPM}$)."

- Finally, we have also moved lines 460 – 466 to line 443, to clarify the distinction between prescribed and equilibrium vPM properties.

- Regarding (4): We assume that equilibrium plume vPM is exclusively comprised of POM and sulphuric acid, with respective assumed densities of 1500 kgm$^{-3}$ and 1800 kgm$^{-3}$, see the caption of Fig. 8. The combined density of the equilibrium plume vPM ($\rho_{vPM}$) is therefore constrained by the relative masses of each component, see the right-hand side of Eq. (22).
- After reviewing (3) and (4), we have decided to revise our initial estimate for the POM density to 1000 kgm$^{-3}$ in line with typical densities of aircraft lubrication oils, and other condensable gaseous emissions including toluene and naphthalene. We have therefore updated Fig. 8 and the following text:

[main text: Fig. 8 caption]

"Figure 8: (a) equilibrium volatile particle modal diameter as a function of fuel sulphur content and the emission index of organic particulate matter, assuming a monomodal distribution and omitting condensation of plume vPM on nvPM, (b) peak plume supersaturation required to activate volatile particles described in (a). We have approximated EI$_{vPM}$ = 1x$10^{17}$, $\varepsilon$ = 3% and $\rho_{vPM}$ as a mass-weighted sum of the densities of sulphuric acid and POM (1800 kgm$^{-3}$ and 15001000 kgm$^{-3}$, respectively), assuming ideal mixing."

Before

[Figure]

[Figure]

After

5. *You perform a sensitivity analysis on the volatile particle emission index by doing 0 or 1e17. Since this is a major hypothesis of your work, I suggest a more detailed sensitivity analysis such as EI=1.e16-18 for example.*

Thank you for your comments. In the original manuscript, we chose to fix $EI_{vPM}$ as we understand that this quantity is bounded by the initial concentration of chemi-ions generated in the exhaust, which are "expected to be relatively invariant" (Yu et al., 1998). Rather, the proportion of these particles that are able to activate is instead dictated by their size and hygroscopicity. However, we acknowledge that because the extended K15 model does not explicitly treat their evolution (other than by dilution), $EI_{vPM}$ may be reduced by coagulation. As suggested, we have emphasised this assumption in our introduction – see (16) below.

In Fig. RR1, we show the impact of changing $EI_{vPM}$. As anticipated, $AEI_{ice}$ is reduced upon reducing $EI_{vPM}$. This is most impactful in the soot-poor regime (left-hand side of Fig. RR1). Notably, the scaling between $AEI_{ice}$ and $EI_{vPM}$ in this regime is non-linear as $AEI_{ice}$ spans two orders of magnitude, while $EI_{vPM}$ spans three orders of magnitude. In the soot-rich regime, increasing $EI_{vPM}$ results in an increased number of ice crystals forming. Overall, similar behaviour could also be achieved through an increase in $GMD_{vPM}$.

[Figure]

Figure. RR1: $AEI_{ice}$ sensitivity to $EI_{nvPM}$ and $EI_{vPM}$.

6. *In the acknowledgement you mention Christiane Voigt for "model comparison with in situ measurement". If you have access to experimental data, why don't you show some experimental validation of your model?*

See response to RC1 (1) and RC2 (2) above.

7. *For what I understand of the model, every volatile particles are a mixture of organics and sulfuric acid. Considering that the organics are mainly insoluble species and sulfuric acid is soluble, is there no possibilities to have in fact two different kind of particles one of pure sulfuric acid and one of pure organics?*

Thank you for your comment. We have addressed this by including the following text in the supplement:

[supplement: SX]

In our model, we assume that all vPM particles have identical chemical composition, which is encompassed by the prescribed total hygroscopicity parameter, $\kappa$. Due to the nature of vapour condensation in the nascent plume, it is possible that the vPM mode may be distributed into several particle types with different chemical composition (Cantin et al., 2024; Yu et al., 1999). To explore the impact of the aforementioned model assumption, we have divided the vPM mode into two distinct modes: vPM-1 and vPM-2, see Table. S1. This is consistent with experimental measurements of chemi-ions, which show that positive chemi-ions (vPM-2) are mostly comprised of protonated organic material (Kiendler et al., 2000b; Sorokin and Arnold, 2006) while negative chemi-ions (vPM-1) are mostly comprised of deprotonated acidic moieties (derivatives of nitric and sulphuric acid) (Kiendler et al., 2000a). As in our original approach, we can effectively modify the chemical composition of vPM-2 by altering the value of $\kappa$ associated with it. Finally, we have assumed that the concentrations of vPM-1 and vPM-2 are equivalent ($EI_{vPM-1} = EI_{vPM-2}$), in line with in-situ observations (Haverkamp et al., 2004).

Table. S1: particle properties for simulations comprising two vPM modes: vPM-1 and vPM-2.

| Mode properties | EI / kg$^{-1}$ | GMD / nm | GSD / no units | $\kappa$ / no units |
|---|---|---|---|---|
| Plume vPM-1 | $5 \times 10^{16}$ | 2.5 | 1.3 | 0.5 ($H_2SO_4$) |
| Plume vPM-2 | $5 \times 10^{16}$ | 2.5 | 1.3 | 0 – 0.5 (organic material) |
| nvPM | $10^{12} – 10^{15}$ | 35 | 2.0 | 0.005 |
| ambient | 600 cm$^{-3}$ | 30 | 2.2 | 0.5 |

[Figure]

**Figure S6: AEI$_{ice}$ sensitivity to EI$_{nvPM}$ and the chemical composition of vPM-2. Note, due to the large increase in the number of tracked size bins when incorporating the additional vPM mode, estimates at large values of EI$_{nvPM}$ (> 10$^{15}$ kg$^{-1}$) could not be obtained.**

**As anticipated, we find that when the composition of vPM-2 is purely sulphuric acid, results are equivalent to assuming a single vPM mode as shown in Fig. 7 of the main text (see Fig. S6). When dividing the vPM mode into a mode comprising sulphuric acid and another comprising organic material, AEI$_{ice}$ estimates depend on the chemical composition of vPM-2. As the hygroscopicity of the vPM-2 mode is reduced, a reduced proportion of these particles activate to form water droplets (and freeze to form ice crystals). Ultimately, when the hygroscopicity of this mode has been reduced below $\kappa$ = 0.01, AEI$_{ice}$ derives exclusively from vPM-1. Therefore, the effective result is a reduction in AEI$_{ice}$. Notably, the effect of dividing the vPM mode in two distinct modes with different values of $\kappa$ is similar to assuming a single mode with a lower mean value of $\kappa$. Therefore, we assert that our sensitivity analysis in Fig. 9 of the main text is sufficient to cover the physical possibility of a more complicated, multicomponent vPM mode. Additionally (as demonstrated in Fig. S6), our model is sufficiently flexible to incorporate additional vPM complexity when the properties of this mode are experimentally grounded.**

This is referred to in the main text as follows:

[main text: line 443]

**"In SI S9 we investigate the impact of assuming all plume vPM particles have the same chemical composition ($\kappa$). To achieve this, we divide the plume vPM mode into two distinct plume vPM modes, prescribing these with identical physical characteristics but dissimilar $\kappa$. We find that the results of the two-mode system are similar to that of the single-mode system with a modified value of $\kappa$. Therefore, we argue that our range of $\kappa$ is sufficient to capture a degree of variation in chemical composition *between* vPM particles."**

**Minor comments**

8. *Line 45-48: You give an experimental definition of volatile particles which is right, however the reader may be confuse if these particles exist or not at the exit of the engine. I suggest adding a sentence, which says that considering the exit temperature of the engine, the volatile particles forms during the cooling of the plume.*

Thank you for this suggestion. We have edited the introduction to clarify that plume vPM forms from condensable gases in the cooling exhaust and that these are therefore gaseous at the engine exit plane.

[main text: lines 46 - 48]

"Aircraft mode particles that evaporate (above) below 350 °C (623 K) (Saffaripour et al., 2020) are defined as (non-)volatile particulate matter (n)vPM. **Engine exhaust exit temperatures typically exceed this threshold, so vPM is thought to form from condensable gases in the cooling plume.** The largest vPM particles are formed from chemi-ions, which are generated during the combustion process (Yu and Turco, 1997)."

9. *Line 50: you give a value for the apparent emission index of volatile particles of 1E17 kg-1. In a recent paper (still a pre-print) https://doi.org/10.21203/rs.3.rs-6559440/v1 they show inflight measurement. The total particle number is limited between 5E14 and 5E15, depending of the flight condition and fuel used. Since this is an order of magnitude lower, it would be great to add more references from the literature.*

Thank you for this suggestion. In the referenced paper (Voigt et al., 2025), the quoted range for the total particle emission index is only for particles strictly larger than 5 nm. This finding is therefore *consistent* with plume vPM properties prescribed in this work. Assuming median plume vPM properties from Table. 1 ($GMD_{vPM}$ = 2.7 nm, $GSD_{vPM}$ = 1.3 and $EI_{vPM}$ = $10^{17}$ kg$^{-1}$), we find projected $EI_{vPM}$ with diameters greater than 5 nm of ~ $10^{15}$ kg$^{-1}$, which is on the same order of magnitude as the values quoted. We have newly incorporated this reference in the main text by reordering this as follows:

[main text: lines 440 - 443]

"For the plume vPM mode, we identify a provisional size range of 1 nm < $GMD_{vPM}$ / nm < 4 and GSD of 1.3 (Kärcher et al., 2000; Yu et al., 1998). Similarly, we assume a fixed emission index $EI_{vPM}$ = $10^{17}$ kg$^{-1}$ in line with modelling and observational evidence (Arnold et al., 2000; Haverkamp et al., 2004; Schumann et al., 2002; Sorokin and Mirabel, 2001). **This is also supported by newer observational evidence (Voigt et al., 2025) that the total particle emission index (particle sizes > 5 nm) is on the order $10^{15}$ kg$^{-1}$, which is consistent with $EI_{vPM}$ = $10^{17}$ kg$^{-1}$ if $GMD_{vPM}$ = 2.7 nm and $GMD_{vPM}$ = 1.3.**"

Moreover, we reason our choice of $EI_{vPM}$ = $10^{17}$ kg$^{-1}$ using four references in line 441 and the caption of Table. 1, which we feel represents sufficient justification.

*10. Line 75-80: I suggest you to read https://doi.org/10.1080/02786826.2024.2395940 and https://doi.org/10.1016/j.aerosci.2025.106612 which introduce in complex CFD microphysical model which seems close to the one from Wong 2014 model Line 226-227: "However, as outline in SIS3, this criterion cannot be reconciled with minimum requirement for particle activation" is a bit too negative since particles may be already big enough to be activated has shown by line 245-251. I suggest to restrict this statement for small particles.*

Thank you for your suggestion. In the referenced papers (Cantin et al., 2024, 2025), the authors use the term "activation" when referring to processes that modify a particle surface, which enables it to *subsequently* uptake water. In this context, an "activated" nvPM particle has adsorbed sulphuric acid and/or water-soluble organic material, that facilitates condensation of water.

This is different to the use of the term "activation" in our manuscript. In our work, we refer to "activation" as the point at which a particle's wet particle diameter exceeds the critical wet particle diameter evaluated using $\kappa$-Köhler theory (Petters and Kreidenweis, 2007), in line with the definition in (Seinfeld and Pandis, 1998). According to this definition, a particle cannot be formally "activated" if the water saturation ratio is less than unity, which is justified in SI S3.

Within our model, the prescription of $\kappa$ = 0.005 for the nvPM is equivalent to assuming that particles have been pre-"activated" (using the earlier (Cantin et al., 2024) definition) by sulphuric acid, which is consistent with a 1% coating by weight (line 464).

*11. Line 270: you neglect the uptake of water of the atmosphere while you take into account the uptake of ambient particle. I agree that at first it is negligible but at the end it will probably influence the size of the ice crystals. Considering the low cost of such modification I don't really understand why have you done this choice.*

Thank you for this suggestion. We recognize that the description of isobaric mixing in Eq. (3) and Fig. 1 implicitly incorporates this effect and therefore we have removed this comment from the original text. In addition, we have corrected the derivation from first principles in Sect. 3.1 and refer the reviewer to the "other changes" section at the end of this document.

*12. In equation 16 you neglect the variation of temperature due to freezing and condensation whereas in the Pyrcel website it is included. Can't it also be included in K15? Moreover in the Pyrcel description in lines 407-412 you replace the gravity term by the one given in this equation, do you keep the latent heats term?*

Thank you for bringing this to our attention. We acknowledge that temperature changes resulting from microphysical processes (particle activation, droplet growth and ice nucleation), should be represented in Eq. 16, when incorporated in pyrcel. Extending this according to (Korolev and Mazin, 2003) gives

$$\frac{dT}{dt} = -\beta \frac{T_{\mathrm{E}} - T_{\mathrm{A}}}{\tau_{\mathrm{m}}} D^{1+\frac{1}{\beta}} + \frac{L_{\mathrm{w}}}{c_{\mathrm{p}}} \frac{dw_{\mathrm{v}}}{dt} + \frac{L_{\mathrm{i}}}{c_{\mathrm{p}}} \frac{dw_{\mathrm{i}}}{dt}, \tag{RR1}$$

where $c_{\mathrm{p}}$ is the isobaric specific heat capacity of moist air, $L_{\mathrm{w/i}}$ and $dw_{\mathrm{v/i}}/dt$ represents the latent heat of water evaporation/ice sublimation and the rate of change of the liquid/ice water mixing ratios, respectively. As you identify, this can be implemented into pyrcel. However, upon implementation, we found that this process only has a limited impact on AEI$_{\mathrm{ice}}$ (< 1%).

Regarding the K15 model, this correction cannot be applied as straightforwardly. This is because there is no assumed feedback between terms $P_w$ and $L_w$ in Eq. 13. However, including $dw_v/dt$ within $P_w$ would couple these terms, undermining the approach to solution.

Note, we use the dilution parameterization provided in (Kärcher et al., 2015), which derives from Figure 4 in (Kärcher, 1999) and in turn (Gerz et al., 1998) (LES simulations behind a Boeing 747). Incidentally, the dilution parameterization provided in (Kärcher et al., 2015) has a similar form to the parameterization presented in (Schumann et al., 1998) (which is used in other microphysical models i.e., (Yu et al., 2024)). This is an empirical result based on a number of *in-situ* observations. Interestingly, several of these observations were obtained from aircraft that were producing contrails, which nevertheless conform to the same dilution profile. This is consistent with our finding using pyrcel, that effects of latent heating are insubstantial for $AEI_{ice}$ estimates.

We have made note of our omission in the main text as follows:

[main text: lines 327 - 329]

"where $\tau_m$ is the timescale over which the contrail mixing parcel is unaffected by entrainment and $\beta$ is a constant dilution parameter. **Note that we choose to omit effects associated with latent heat within our simulations, which we found to have a negligible impact on projected ice crystal number concentrations.**"

> 13. *In line 450-453 you distinguish between oil organics and gaseous emission. In line 461 I have the impression that you treat them as the same species. Are they treated the same way?*

Yes, this is correct. In lines 450 – 453 we differentiate between organic emissions by qualifying whether they derive from "combustion" or "non-combustion" sources. To that end, we regard lubrication oil as "non-combustion" organic material in line with (Timko et al., 2014). The remaining organic material necessarily derives from "combustion" processes. Ultimately, both sources are combined to estimate the total mass emission index of "condensable [organic] gaseous emissions". This quantity is used as an approximate upper bound in Fig. 8.

> 14. *You conclude 4.3 by telling that Pyrcel is better than K15 but in line 554 you tell that you make the sensitivity analysis with K15. I guess you have use Pyrcel but it is just a typo.*

Thank you for noticing this typo. We have changed this in the text as follows:

[main text: lines 554 - 555]

"An evaluation of the **modified pyrcel**  model sensitivity across a range of $T_A$ and $EI_{nvPM}$ revealed several key findings (Fig. 7)."

15. *In line 645 you say that the K15 model could be incorporated in global contrail simulations, however considering the comparison with Pyrcel, I wonder why K15 and not Pyrcel?*

Thank you for your comment. You are right to say that, in principle, either model could be used to generate lookup tables. In practice, the modified pyrcel model requires approximately 100 times greater computational effort per simulation, which is why we mentioned the extended K15 model in this instance. However, as we acknowledge that either model could be used in principle, we have decided to make the following change:

[main text: lines 679 - 681]

"While the extended K15 and modified pyrcel models are both too computationally demanding to be directly integrated into global contrail simulations, we suggest that **either** models could be incorporated in these simulations by preparing multidimensional lookup tables."

16. *In Supplementary material S1, figure S1: please recall the k value used in order to simplify the reading.*

Thank you for this suggestion. We have introduced the $\kappa$ value in the caption of Fig. S1 as shown below:

[supplement: Fig. S1 caption]

"Figure S1: illustrative ice nucleation temperature as predicted using $J$ parametrizations from (Koop et al., 2000) using the methodology outlined in (Kärcher et al., 2015). Particle properties have been taken from Table. 1, assuming a sulphur-rich vPM mode **($\kappa$ = 0.5)** with $GMD_{vPM}$ = 4 nm. Several contrail mixing lines have also been presented with $G$ = 1.64."

17. *Limitation given in line 351-354 has to be emphasis in the introduction since it is an important one.*

Thank you for this suggestion. We agree that this is the fundamental limitation of our study and have now addressed this clearly in the introduction:

[main text: lines 86 - 88]

"Here, we undertake a literature review of the microphysical pathway to contrail formation, to better understand the role of plume vPM in contrail formation. We then extend two parcel models using detailed microphysics to account for activation of plume vPM. **In both models, we prescribe plume vPM properties at the time of droplet formation and ice nucleation (0.1 – 1 s after emission) rather than explicitly modelling their formation.** The **two** models include: (i) a minimal microphysical framework developed in (Kärcher et al., 2015), henceforth referred to using the shorthand K15 and (ii) a more complex numerical parcel model (pyrcel) developed in (Rothenberg and Wang, 2016)."

18. *In line 54 of the supplementary material, you give the probability to transform into ice. Then in line 56 you say that the particle freeze when the probability is one. However, the line 54 formula shows that this probability is never equal to one. Then you have to choose a threshold. You have to give the information to the reader.*

Thank you for noticing this. Strictly, we assume that $\lambda = 1 - \delta$, where $\delta = 10^{-20} \sim 0$. We have made this change in the supplement text.

[supplement: lines 56 - 57]

For the below analysis, we choose $\lambda = 1 - \delta$ ($\delta = 10^{-20} \sim 0$) in line with previous studies (Kärcher et al., 2015; Lewellen, 2020) and evaluate $r_{\text{frz}}^{-1}$ according to Eq. (S2).

**Other changes**

1. Updated governing parcel model relations:

After reviewing Sect. 3.1, we have noticed an error in the original derivation and have addressed this as follows:

[main text: lines 259 – 275]

As discussed in Sect. 2.2, the critical parameter governing uptake of water onto particles is the plume supersaturation, $S_v$. The supersaturation within a parcel of air can be described as

$$S_v = \frac{P_T}{p_{\text{liq}}(\varepsilon + w_v)} w_v,$$ (10)

where $\varepsilon$ (= 0.622) is the ratio of the molar mass of water (18.02 gmol$^{-1}$) to the molar mass of dry air (28.97 gmol$^{-1}$) and $w_v$ is the water vapour mixing ratio, the mass of water vapour per unit mass of dry air contained within the parcel. **We can simplify Eq. 10 further using several features of contrail mixing. Within a contrail, the maximum partial pressure of water vapour in the plume is given by the maximum of $S_v\, p_{\text{liq}}$. This is bounded by $S_{v,M}\, p_{\text{liq}}$, since isobaric mixing is the only process that acts to increase the parcel saturation ratio. Assuming that the ambient environment is ice-*saturated*, we also know that $p_{v,M}$ is bounded by $(p_{v,M})_{\text{max}} = p_{\text{ice}}\,(T_{\text{SAC}}) + G\,(T_E - T_{\text{SAC}})$. Rearranging Eq. (10), we can therefore bound $w_v$ as**

$$w_v < \frac{\varepsilon(p_{v,M})_{\text{max}}}{P_T - (p_{v,M})_{\text{max}}}.$$ (11)

Using $T_E$ = 600 K, $T_{\text{SAC}}$ = 224 K, $G$ = 1.64 PaK$^{-1}$ and $P_T$ = 23000 Pa, we have that $w_v < 0.02$. Therefore, the inequality $w_v \ll \varepsilon$ is satisfied for all conditions, which enables us to simplify Eq. (10) as

$$S_v = \frac{P_T}{p_{\text{liq}}\varepsilon} w_v.$$ (12)

Next, we assume that contrail mixing occurs at constant atmospheric pressure so that upon differentiating Eq. (12) with respect to time and collecting like-terms, we have

$$\frac{dS_v}{dt} = \frac{P_T}{p_{liq}\varepsilon}\frac{dw_v}{dt} - S_v\frac{1}{p_{liq}}\frac{dp_{liq}}{dt},$$ (13)

$$\frac{dw_v}{dt} = -\left(\frac{dw_c}{dt}\right)_{microphysical} + \left(\frac{dw_{v,M}}{dt}\right)_{mixing}.$$ (14)

Equation (14) is a statement of mass conservation: **any change in the water vapour content of the parcel must result from either particle microphysical processes (i.e., particle activation and droplet/ice crystal growth) or from plume mixing, provided there is no entrainment of ambient water vapour.** ~~any change in the water vapour content of the parcel must have an equal, but opposite effect on the condensed liquid content, (dw_c)/dt, provided there is no entrainment. Although we will later explicitly describe entrainment of ambient aerosol within the mixing contrail plume, we assume that entrainment of ambient tropospheric water vapour does not impact the contrail plume evolution. This is because the plume vapour mixing ratio at the plateau in particle activation is typically several orders of magnitude larger than background levels. Therefore, the entrainment of small quantities of ambient tropospheric air is assumed to have a negligible impact on the final droplet number concentration, and as such, this process is omitted from Eq. (11). We can generalize Eq. (11) in the form~~ **Using Eq. (10) we can express the mixing term as**

$$\left(\frac{dw_{v,M}}{dt}\right)_{mixing} = \dot{T}\frac{d}{dT}\left(\frac{p_{v,M}\varepsilon}{P_T}\right) = \frac{\varepsilon}{P_T}G\dot{T}.$$ (15)

**Then, combining Eq. (12) – (15), we arrive at the governing equation**

$$\frac{dS_v}{dt} = \frac{1}{p_{liq}}G\dot{T} - S_v\frac{1}{p_{liq}}\frac{dp_{liq}}{dt} - \frac{P_T}{p_{liq}\varepsilon}\frac{dw_c}{dt}.$$ (16)

**This is almost identical to the result given in (Kärcher et al., 2015). However, we find that on the right-hand side of Eq. (16), the actual plume saturation ratio ($S_v$) is present rather than the saturation ratio assumed from mixing alone ($S_{v,M}$). This equation can be solved numerically, for example by using a numerical parcel model. However, the equation cannot be solved analytically. Under these circumstances, a solution may be derived by approximating $S_v \sim S_{v,M}$ on the right-hand side of Eq. (16) as in the original K15 model, which effectively decouples the equation. In this purely analytical case, we can generalize Eq. (16) in the form**

$$\frac{dS_v}{dt} = P_w - L_w,$$ (17)

where $P_w$ represents the rate of change of supersaturation without particles in the plume, and $L_w$ represents the rate of change of supersaturation resulting from (a) particle activation to form water droplets and (b) growth of droplets and/or ice crystals. **The** description for $P_w$ **follows** from the description of contrail mixing outlined in Sect. 2.1 and is given by (Kärcher et al., 2015)

$$P_w = \frac{1}{p_{liq}}G\dot{T} - S_{v,M}\frac{1}{p_{liq}}\frac{dp_{liq}}{dt} = \frac{dS_{v,M}}{dt} = \frac{d}{dT}\left(\frac{p_{v,M}}{p_{liq}}\right)\frac{dT}{dt},$$ (18)

where  $\dot{T}$ is the cooling rate in the plume.

2. Correcting notation:

[main text: Eq. (20)]

$$L_{\text{w}} = \frac{P_T}{p_{\text{liq}} e_s^{\theta} \varepsilon} \frac{dw_c}{dt} \; ; \quad \frac{dw_c}{dt} = \frac{4\pi\rho_w}{\rho_a} \sum_{i=1}^{n} n_{w,i} r_{w,i}^2 \dot{r}_{w,i},$$ **(24)**

3. Changes to code:

Correction made to description $dw_v/dt$ in the pyrcel model code. This has a negligible effect on simulation outputs.

4. Uncertainty estimate

When reproducing some of the figures in the original manuscript, we noticed that pyrcel $AEI_{\text{ice}}$ outputs were sensitive to the prescribed number of size bins used for a given mode. Therefore, we used this sensitivity to estimate the uncertainty on $AEI_{\text{ice}}$ estimates, which we have introduced to the supplement. We have also combined this with S11 (line 414 onwards of the supplement).

[supplement: SI SX]

"**In pyrcel, changing the number of size bins for a given mode changes the discretization of particle diameters. Given the parameterization for activation is highly sensitive to particle diameter (see Eq. (9) in the main text), small adjustments to these values can therefore impact activated particle number concentrations and $AEI_{\text{ice}}$. During our analyses, we noticed that $AEI_{\text{ice}}$ was more sensitive to the prescribed number of size bins for vPM than for nvPM. We suggest that this is likely a consequence of two factors. Firstly, the GMD of the vPM particle size distribution is smaller than for the nvPM particle size distribution (see Table. 1 in the main text) and activated number concentrations are increasingly sensitive to smaller particle diameters (see Fig. 2b in the main text). Secondly, the droplet and ice (see Fig. S7) particle size distributions for nvPM and vPM have different forms. For nvPM, the maximum in the ice particle size distribution is distinctly separated from the minimum diameter of activation. However, for vPM, the maximum in the ice particle size distribution *is* equivalent to the minimum diameter of activation. As a result, small changes in the quantization of size bins (that modify the minimum diameter of activation) have a large effect on the number concentration of vPM-derived ice crystals but not on the concentration of nvPM-derived ice crystals.**

**For this reason, we decided to interrogate model sensitivity to the prescribed number of (n)vPM size bins. We chose to investigate the sensitivity under soot-poor conditions ($EI_{\text{nvPM}} = 10^{12}$ kg$^{-1}$) as this represents the point at which $AEI_{\text{ice}}$ is maximally sensitive to vPM. Therefore, we can use the sensitivity at this point to derive maximum uncertainties. Accordingly, simulations were performed using a sulphur-rich plume vPM mode (according to Table. 1 in the main text), $T_A$ = 215 K and $EI_{\text{nvPM}} = 10^{12}$ kg$^{-1}$. When varying the number of vPM size bins, other size bin numbers were fixed at 50. Model sensitivity to the number of vPM size bins is shown in Fig. S8.**

[Figure]

**Figure S7: particle size distributions for (a) nvPM and (b) vPM illustrative of typical contrail mixing behaviour. The particle size distributions have been shown at four different temperatures during plume evolution and the phase (solid/liquid aerosol, liquid droplet, ice crystal) of each bin has been indicated using a different marker.**

[Figure]

**Figure S8: AEI$_{ice}$ sensitivity (normalised by EI$_{nvPM}$ = 10$^{12}$ kg$^{-1}$) to the number of size bins prescribed for vPM. An envelope is drawn to constrain maximum errors associated with each vPM bin number.**

**Previously, we found that AEI$_{ice}$ estimates were largely insensitive to the number of prescribed nvPM size bins. However, for vPM, we find that AEI$_{ice}$ estimates converge with increasing numbers of size bins. By enveloping these estimates, we find that errors range from -10% to +9% for 150 size bins, which is the typical number used in our model simulations. Therefore, this represents the maximum uncertainty in our AEI$_{ice}$ estimates."**

Considering this analysis, we have made the following changes in the main text and figures:

[main text: lines 522 – 523]

"When plume vPM is included, we find that under this range of EI$_{nvPM}$, differences in model predictions for AEI$_{ice}$ and $s_{v, max}$ exhibit similar trends, see Fig. 5b. **Note that AEI$_{ice}$ predictions made using the modified pyrcel model were found to have a maximum uncertainty of ± 10%, see SI S10. These derive principally from discretizing the vPM mode, which necessarily approximates the analytic particle size distribution, see SI S4. These errors are shown in Fig. 6 and future analysis, where appropriate."**

[main text: Fig. 7]

[Figure]

[main text: Fig. 6]

[Figure]

[main text: line 14]

"We find model agreement within 10-30% in the previously defined "soot-poor" regime."

[main text: lines 708 - 709]

"However, when plume vPM activation is included, the modified pyrcel model predicts higher ice crystal number concentrations by ~10-30% for $EI_{nvPM} \leq 10^{13}$ kg$^{-1}$.

[main text: lines 521-522]

"Without plume vPM, we find the model agreement for AElice is within ~35% when $EI_{nvPM}$ < $10^{16}$ kg$^{-1}$, see Fig. 6a."

[main text: lines 707-708]

"Generally, the estimated AElice from both models are within 35% for $EI_{nvPM} \leq 10^{16}$ kg$^{-1}$, when activation of plume vPM is excluded"

5.  Correcting statement

After rereading the original manuscript, we became aware that the below clause was not strictly true and have since removed it.

[main text: lines 157-158]

"Importantly, , the SAC implies that the activation of particles into liquid droplets occurs before ice nucleation takes place."

**References**

Arnold, F., Kiendler, A., Wiedemer, V., Aberle, S., Stilp, T., and Busen, R.: Chemiion concentration measurements in jet engine exhaust at the ground: Implications for ion chemistry and aerosol formation in the wake of a jet aircraft, Geophysical Research Letters, 27, 1723–1726, https://doi.org/10.1029/1999GL011096, 2000.

Bier, A. and Burkhardt, U.: Impact of Parametrizing Microphysical Processes in the Jet and Vortex Phase on Contrail Cirrus Properties and Radiative Forcing, JGR Atmospheres, 127, e2022JD036677, https://doi.org/10.1029/2022JD036677, 2022.

Burkhardt, U., Bock, L., and Bier, A.: Mitigating the contrail cirrus climate impact by reducing aircraft soot number emissions, npj Clim Atmos Sci, 1, 37, https://doi.org/10.1038/s41612-018-0046-4, 2018.

Cantin, S., Chouak, M., and Garnier, F.: Eulerian–Lagrangian CFD-microphysics modeling of aircraft-emitted aerosol formation at ground-level, Aerosol Science and Technology, 58, 1347–1370, https://doi.org/10.1080/02786826.2024.2395940, 2024.

Cantin, S., Chouak, M., and Garnier, F.: Effects of Fuel Sulfur Content and Nvpm Emissions on Contrail Formation: A Cfd-Microphysics Study Including the Role of Organic Compounds, https://doi.org/10.2139/ssrn.5138455, 2025.

Chen, C.-C. and Gettelman, A.: Simulated radiative forcing from contrails and contrail cirrus, Atmos. Chem. Phys., 13, 12525–12536, https://doi.org/10.5194/acp-13-12525-2013, 2013.

Chen, C.-C. and Gettelman, A.: Simulated 2050 aviation radiative forcing from contrails and aerosols, Atmos. Chem. Phys., 16, 7317–7333, https://doi.org/10.5194/acp-16-7317-2016, 2016.

Gerz, T., Dürbeck, T., and Konopka, P.: Transport and effective diffusion of aircraft emissions, J. Geophys. Res., 103, 25905–25913, https://doi.org/10.1029/98jd02282, 1998.

Haverkamp, H., Wilhelm, S., Sorokin, A., and Arnold, F.: Positive and negative ion measurements in jet aircraft engine exhaust: concentrations, sizes and implications for aerosol formation, Atmospheric Environment, 38, 2879–2884, https://doi.org/10.1016/j.atmosenv.2004.02.028, 2004.

Jones, S. H. and Miake-Lye, R. C.: Parameterization of $H_2 SO_4$ and organic contributions to volatile PM in aircraft plumes at ground idle, Journal of the Air & Waste Management Association, 74, 490–510, https://doi.org/10.1080/10962247.2024.2354820, 2024.

Kärcher, B.: Aviation-Produced Aerosols and Contrails, Surveys in Geophysics, 20, 113–167, https://doi.org/10.1023/a:1006600107117, 1999.

Kärcher, B., Turco, R. P., Yu, F., Danilin, M. Y., Weisenstein, D. K., Miake-Lye, R. C., and Busen, R.: A unified model for ultrafine aircraft particle emissions, J. Geophys. Res., 105, 29379–29386, https://doi.org/10.1029/2000JD900531, 2000.

Kärcher, B., Burkhardt, U., Bier, A., Bock, L., and Ford, I. J.: The microphysical pathway to contrail formation, JGR Atmospheres, 120, 7893–7927, https://doi.org/10.1002/2015JD023491, 2015.

Kiendler, A., Aberle, S., and Arnold, F.: Negative chemiions formed in jet fuel combustion: new insights from jet engine and laboratory measurements using a quadrupole ion trap mass spectrometer apparatus, Atmospheric Environment, 34, 2623–2632, https://doi.org/10.1016/s1352-2310(99)00475-6, 2000a.

Kiendler, A., Aberle, S., and Arnold, F.: Positive ion chemistry in the exhaust plumes of an air craft jet engine and a burner: investigations with a quadrupole ion trap mass spectrometer, Atmospheric Environment, 34, 4787–4793, https://doi.org/10.1016/s1352-2310(00)00253-3, 2000b.

Koop, T., Luo, B., Tsias, A., and Peter, T.: Water activity as the determinant for homogeneous ice nucleation in aqueous solutions, Nature, 406, 611–614, https://doi.org/10.1038/35020537, 2000.

Korolev, A. V. and Mazin, I. P.: Supersaturation of Water Vapor in Clouds, Journal of the Atmospheric Sciences, 60, 2957–2974, https://doi.org/10.1175/1520-0469(2003)060<2957:SOWVIC>2.0.CO;2, 2003.

Lewellen, D. C.: A Large-Eddy Simulation Study of Contrail Ice Number Formation, Journal of the Atmospheric Sciences, 77, 2585–2604, https://doi.org/10.1175/JAS-D-19-0322.1, 2020.

Petters, M. D. and Kreidenweis, S. M.: A single parameter representation of hygroscopic growth and cloud condensation nucleus activity, Atmospheric Chemistry and Physics, 7, 1961–1971, https://doi.org/10.5194/acp-7-1961-2007, 2007.

Rojo, C., Vancassel, X., Mirabel, P., Ponche, J.-L., and Garnier, F.: Impact of alternative jet fuels on aircraft-induced aerosols, Fuel, 144, 335–341, https://doi.org/10.1016/j.fuel.2014.12.021, 2015.

Rothenberg, D. and Wang, C.: Metamodeling of Droplet Activation for Global Climate Models, Journal of the Atmospheric Sciences, 73, 1255–1272, https://doi.org/10.1175/JAS-D-15-0223.1, 2016.

Saffaripour, M., Thomson, K. A., Smallwood, G. J., and Lobo, P.: A review on the morphological properties of non-volatile particulate matter emissions from aircraft turbine engines, Journal of Aerosol Science, 139, 105467, https://doi.org/10.1016/j.jaerosci.2019.105467, 2020.

Schumann, U., Schlager, H., Arnold, F., Baumann, R., Haschberger, P., and Klemm, O.: Dilution of aircraft exhaust plumes at cruise altitudes, Atmospheric Environment, 32, 3097–3103, https://doi.org/10.1016/s1352-2310(97)00455-x, 1998.

Schumann, U., Arnold, F., Busen, R., Curtius, J., Kärcher, B., Kiendler, A., Petzold, A., Schlager, H., Schröder, F., and Wohlfrom, K.-H.: Influence of fuel sulfur on the composition of aircraft exhaust plumes: The experiments SULFUR 1–7, Journal of Geophysical Research: Atmospheres, 107, AAC 2-1-AAC 2-27, https://doi.org/10.1029/2001JD000813, 2002.

Seinfeld, J. H. and Pandis, S. N.: Atmospheric chemistry and physics: from air pollution to climate change, Wiley, New York, 1326 pp., 1998.

Sorokin, A. and Arnold, F.: Organic positive ions in aircraft gas-turbine engine exhaust, Atmospheric Environment, 40, 6077–6087, https://doi.org/10.1016/j.atmosenv.2006.05.038, 2006.

Sorokin, A. and Mirabel, P.: Ion recombination in aircraft exhaust plumes, Geophysical Research Letters, 28, 955–958, https://doi.org/10.1029/2000GL011801, 2001.

Teoh, R.: Impact of Volatile Particulate Matter on Global Contrail Radiative Forcing and Mitigation Assessments.

Teoh, R., Engberg, Z., Schumann, U., Voigt, C., Shapiro, M., Rohs, S., and Stettler, M. E. J.: Global aviation contrail climate effects from 2019 to 2021, Atmos. Chem. Phys., 24, 6071–6093, https://doi.org/10.5194/acp-24-6071-2024, 2024.

Timko, M. T., Albo ,Simon E., Onasch ,Timothy B., Fortner ,Edward C., Yu ,Zhenhong, Miake-Lye ,Richard C., Canagaratna ,Manjula R., Ng ,Nga Lee, and and Worsnop, D. R.: Composition and Sources of the Organic Particle Emissions from Aircraft Engines, Aerosol Science and Technology, 48, 61–73, https://doi.org/10.1080/02786826.2013.857758, 2014.

Voigt, C., Märkl, R., Sauer, D., Dischl, R., Kaufmann, S., Bräuer, T., Jurkat, T., Renard, C., Seeliger, K., Chenadec, G. L., Moreau, J., Yu, F., Bonne, N., Roche, A., Zelina, J., Dörnbrack, A., Eirenschmalz, L., Heckl, C., Horst, E., Lichtenstern, M., Marsing, A., Roiger, A., Scheibe, M., Stock, P., Giez, A., Eckel, G., Neumann, G., Vals, M., Requena-Esteban, E., and Clercq, P. L.: Substantial aircraft contrail formation at low soot emission levels, https://doi.org/10.21203/rs.3.rs-6559440/v1, 9 May 2025.

Wong, H.-W., Jun, M., Peck, J., Waitz, I. A., and Miake-Lye, R. C.: Detailed Microphysical Modeling of the Formation of Organic and Sulfuric Acid Coatings on Aircraft Emitted Soot Particles in the Near Field, Aerosol Science and Technology, 48, 981–995, https://doi.org/10.1080/02786826.2014.953243, 2014.

Wong, H.-W., Jun, M., Peck, J., Waitz, I. A., and Miake-Lye, R. C.: Roles of Organic Emissions in the Formation of Near Field Aircraft-Emitted Volatile Particulate Matter: A Kinetic Microphysical Modeling Study, Journal of Engineering for Gas Turbines and Power, 137, 072606, https://doi.org/10.1115/1.4029366, 2015.

Yu, F. and Turco, R. P.: The role of ions in the formation and evolution of particles in aircraft plumes, Geophysical Research Letters, 24, 1927–1930, https://doi.org/10.1029/97GL01822, 1997.

Yu, F., Turco, R. P., Kärcher, B., and Schröder, F. P.: On the mechanisms controlling the formation and properties of volatile particles in aircraft wakes, Geophysical Research Letters, 25, 3839–3842, https://doi.org/10.1029/1998GL900012, 1998.

Yu, F., Turco, R. P., and Kärcher, B.: The possible role of organics in the formation and evolution of ultrafine aircraft particles, J. Geophys. Res., 104, 4079–4087, https://doi.org/10.1029/1998JD200062, 1999.

Yu, F., Kärcher, B., and Anderson, B. E.: Revisiting Contrail Ice Formation: Impact of Primary Soot Particle Sizes and Contribution of Volatile Particles, Environ. Sci. Technol., 58, 17650–17660, https://doi.org/10.1021/acs.est.4c04340, 2024.